# Bacterial persistence is an active σ<sup>S</sup> stress response to metabolic flux limitation

Jakub Leszek Radzikowski[1], Silke Vedelaar[1], David Siegel[2], Álvaro Dario Ortega[1], Alexander Schmidt[3] & Matthias Heinemann[1,*]

## Abstract

While persisters are a health threat due to their transient antibiotic tolerance, little is known about their phenotype and what actually causes persistence. Using a new method for persister generation and high-throughput methods, we comprehensively mapped the molecular phenotype of *Escherichia coli* during the entry and in the state of persistence in nutrient-rich conditions. The persister proteome is characterized by σ<sup>S</sup>-mediated stress response and a shift to catabolism, a proteome that starved cells tried to but could not reach due to absence of a carbon and energy source. Metabolism of persisters is geared toward energy production, with depleted metabolite pools. We developed and experimentally verified a model, in which persistence is established through a system-level feedback: Strong perturbations of metabolic homeostasis cause metabolic fluxes to collapse, prohibiting adjustments toward restoring homeostasis. This vicious cycle is stabilized and modulated by high ppGpp levels, toxin/anti-toxin systems, and the σ<sup>S</sup>-mediated stress response. Our system-level model consistently integrates past findings with our new data, thereby providing an important basis for future research on persisters.

**Keywords** *Escherichia coli*; metabolism; persistence; proteomics; stress response

**Subject Categories** Metabolism; Microbiology, Virology & Host Pathogen Interaction; Quantitative Biology & Dynamical Systems

**Mol Syst Biol. (2016) 12: 882**

## Introduction

Bacterial persistence is a phenotypic state of transient antibiotic tolerance that threatens human and animal health (Cohen *et al*, 2013; Grant & Hung, 2013). This state is typically associated with dormancy in nutrient-rich environments and with absent or low antibiotic target activity, which renders most antibiotics ineffective (Lewis, 2010) and can cause recurrence of infections with, for example, *Mycobacterium, Staphylococcus,* or *Pseudomonas* species (Dawson *et al*, 2011; Fauvart *et al*, 2011; Cohen *et al*, 2013). Despite the importance of persisters, we still have very limited insights into the molecular phenotype of these cells and into what actually triggers persistence.

Bacterial persistence is typically investigated using different *in vitro* models. One persistence model are the antibiotic-tolerant cells that are formed stochastically in growing cultures (Maisonneuve *et al*, 2013; Feng *et al*, 2014). Another model for persistence are starved cells (i.e. cells in stationary phase) (Nguyen *et al*, 2011), which have diminished or absent antibiotic target activity due to the absence of nutrients (Fung *et al*, 2010). However, it is questionable whether this model is relevant for all types of persisters, as they occur in the host both in the presence (Guarner & Malagelada, 2003; Rohmer *et al*, 2011) or absence of nutrients (Appelberg, 2006). Finally, a third model for persistence was recently proposed: It was found that after certain nutrient shifts (i.e. abrupt shifts or gradual shifts resembling diauxie) a large number of non-/slow-growing and antibiotic-tolerant cells (i.e. persisters) emerge in nutrient-rich conditions (Amato & Brynildsen, 2014; Kotte *et al*, 2014).

Through isolation of stochastically generated persisters from growing cultures by means of ampicillin treatment or FACS and through performing transcriptome analyses, it was found that—compared to growing cells—persisters have higher abundances of SOS response, cold/hot shock, and toxin/antitoxin systems (TAS) as well as lower levels of flagellum-related transcripts (Keren *et al*, 2004; Shah *et al*, 2006). Further, it was found that the fraction of such persister cells could be increased by overexpression of certain toxins such as HipA (Korch & Hill, 2006), RelE (Tashiro *et al*, 2012), YgiU (Shah *et al*, 2006), or the Lon protease (Maisonneuve *et al*, 2011). On the basis of these findings, it was suggested that toxins, higher expressed in individual cells (eventually due to stochastic variation in ppGpp levels), are the decisive factor for persister formation (Maisonneuve *et al*, 2013). However, neither ppGpp-negative strains (Maisonneuve *et al*, 2013), rpoS deletion strains (Nguyen *et al*, 2011), nor strains with multiple TAS deleted (Maisonneuve *et al*, 2011) were completely free of persisters,

1   Molecular Systems Biology, Groningen Biomolecular Sciences and Biotechnology Institute, University of Groningen, Groningen, The Netherlands
2   Analytical Biochemistry, Groningen Research Institute of Pharmacy, University of Groningen, Groningen, The Netherlands
3   Biozentrum, University of Basel, Basel, Switzerland
    *Corresponding author. Tel: +31 50 363 8146; E-mail: m.heinemann@rug.nl

suggesting that persister formation can also be achieved through other mechanisms. The involvement of various mechanisms could explain the recently observed heterogeneity between persister cells (Amato & Brynildsen, 2015).

In fact, all three persister models indicate that persister formation could also involve metabolism; next to the involvement of toxin/antitoxin systems. First, the observation that the frequency of stochastically formed persisters increased with the amount of glucose transport inhibitor added to the growth medium (Maisonneuve et al, 2013) shows that stochastic persister formation depends, at least to some degree, on the magnitude of metabolic flux in single cells. Second, for the nutrient-shift-induced persisters, we demonstrated that a limitation in metabolic flux is decisive whether an individual cell adapts to the new nutrient or enters the persister state (Kotte et al, 2014). Finally, in the starvation model, lack of nutrients (and thus lack of metabolic flux) is the cause for the observed persister phenotype. Together, these findings suggest that the metabolic state of a cell and persistence might be closely tethered.

Still, our knowledge about persisters and what exactly triggers their formation is limited. The main problem is that cultures containing only a small fraction of persister cells cannot be subjected to population-averaging experimental methods that require large number of cells. As a matter of fact, the proteome, metabolome, and the physiology (i.e. growth and nutrient uptake) of persister cells are not yet known, as recently highlighted (Balaban et al, 2013; Amato et al, 2014). Here, however, the discovery of the nutrient-shift-induced persisters opened up a new possibility to investigate this important bacterial phenotype.

In this work, we exploited the nutrient-shift method to generate large numbers of persisters present in nutrient-rich environments. First, we demonstrated similarities of these persisters with the stochastically generated ones in terms of antibiotic tolerance, TAS upregulation, and ppGpp levels. Then, we determined the phenotype of nutrient-shift-induced and starvation-induced persisters, including the proteome, metabolite levels and physiology of cells during the entry into and in the state of persistence, thereby covering at least two of the currently used persister models. This comparative analysis allowed us to determine the influence of nutrient presence. We found that the metabolism of persisters formed during abrupt glucose-to-fumarate shift is characterized by low carbon source uptake that sustains a metabolism sufficient for ATP maintenance requirements, and slow growth. Further, we found the proteome of these persisters to be shaped by $\sigma^S$, typically associated with starvation and stress. On the basis of our data and previous knowledge, we developed a system-level model on the emergence and sustenance of persisters, which we validated through a series of targeted perturbation experiments. The generated comprehensive description of the persister phenotype and the developed model will form an important basis for future work toward understanding and eradicating persisters.

## Results

Following our previous work (Kotte et al, 2014), when we switched *Escherichia coli* cells from glucose to fumarate medium, only an extremely small fraction of cells ($0.1 \pm 0.05\%$, SD) adapted and started to grow on fumarate. The other cells, despite the presence of a utilizable carbon source, entered a state of non-/slow growth (Kotte et al, 2014), resembling the one of persister cells. Because even 10–15 h after the nutrient shift the growing population reached only 1% of the total population (Appendix Fig S1A), we could perform population-averaging proteome analyses, metabolite concentration measurements, and physiological analyses, with the small fraction of growing cells not significantly influencing the results in the first 8 h after the shift (cf. Appendix Text S1). We performed the same analyses on cells that we switched from glucose to medium without a carbon source, generating starved cells, which allowed us to investigate the effect of nutrient presence on the persister phenotype.

### Nutrient shifts induce persistence

First, we asked whether the non-/slow-growing cells in nutrient-rich conditions obtained after a nutrient shift and starved cells resemble stochastically induced persister cells in growing cultures. With the key hallmark of persister cells being transient antibiotic tolerance (Balaban et al, 2013), we specifically asked whether and how fast the cells become tolerant to antibiotics after the switch to fumarate or to no carbon source. Therefore, we treated the cells for 2 h with ampicillin at a concentration that was used for detection of stochastically formed persisters (Maisonneuve et al, 2011) and that killed fumarate-growing cells, at different time points after the switch to fumarate medium or medium without a carbon source, and then determined the fraction of surviving cells. In both cases, we found that virtually all cells became tolerant to ampicillin within 30 min after the nutrient shift (Fig 1A).

To determine whether the cells also became tolerant against other antibiotics and to identify the dependence of this tolerance on nutrient presence/absence, we exposed the starved cells and the non-/slow-growing cells in nutrient-rich conditions (i.e. in the presence of fumarate as the carbon source) 4 h after the medium shift, to six different antibiotics for 2 h, at concentrations that killed fumarate-growing cells, and determined the fractions of surviving cells. Choosing different antibiotics was motivated by their different mechanisms of action. Observed differences could hint to specific mechanisms increasing antibiotic tolerance between the two tested persister models. Here, we found tolerant cells for all tested antibiotics (Fig 1B). For CCCP (proton gradient disruptor), we found that the non-/slow-growing cells in nutrient-rich conditions survived the antibiotic challenge significantly better (t-test, P-value < 0.01) compared to the starved cells. Our results show that non-/slow-growing cells in nutrient-rich conditions and starved cells are tolerant to numerous antibiotics at concentrations that killed fumarate-growing cells. However, the observed difference—namely, the higher tolerance of non-/slow-growing cells in nutrient-rich conditions against CCCP compared to starved cells—suggested that persister cells in nutrient-rich conditions must exploit specific tolerance mechanism that enhance their survival over that of persisters generated by starvation.

Elevated levels of ppGpp, occurring for example during amino acid starvation, have been associated with the persister phenotype (Amato et al, 2013; Maisonneuve et al, 2013). To determine whether ppGpp levels are also increased in the non-/slow-growing cells obtained after the nutrient shift, we quantified the intracellular ppGpp concentration in these cells, as well as in cells exponentially

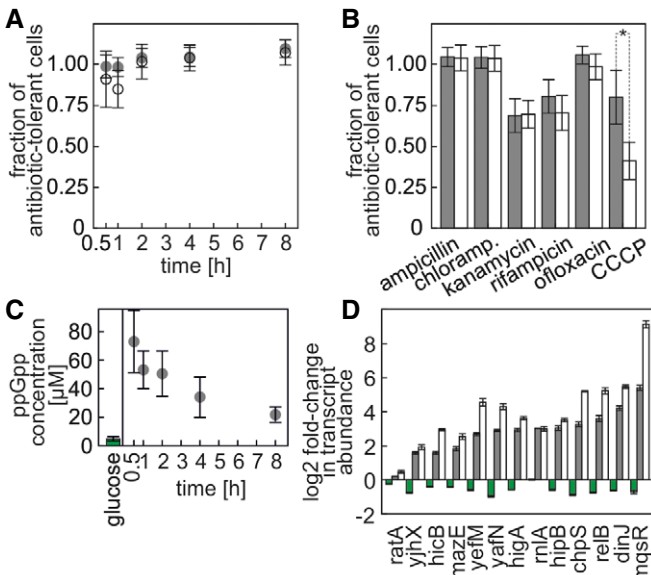

**Figure 1. Non-/slow-growing cells and starved cells are antibiotic-tolerant, accumulate ppGpp, and express TAS.**

A  Dynamics of establishing antibiotic tolerance during entry into non-/slow growth or starvation. Fraction of antibiotic-tolerant cells after treatment with ampicillin (2 h, 100 μg ml⁻¹) is shown at various times after the medium switch. Gray disks: non-/slow-growing cells, open circles: starved cells. Data from biological triplicate. Error bars represent one standard deviation.

B  Fractions of antibiotic-tolerant cells after a 2-h treatment of non-/slow-growing and starved cells with various antibiotics (ampicillin 100 μg ml⁻¹; chloramphenicol 140 μg ml⁻¹; kanamycin 100 μg ml⁻¹; ofloxacin 5 μg ml⁻¹; rifampicin 100 μg ml⁻¹; CCCP 50 μg ml⁻¹) 4 h after nutrient switch. Gray bars: non-/slow-growing cells, white bars: starved cells. Data from biological triplicate. Error bars represent one standard deviation. Statistical significance (t-test or Wilcoxon rank sum test for kanamycin and ofloxacin, P-value < 0.05) is marked with *.

C  ppGpp concentration in cells growing on glucose and in cells shifted from glucose-to-fumarate medium. Data from biological triplicate. Error bars represent one standard deviation.

D  Log2 fold change in transcript abundance of first genes in TAS operons compared to cells growing on glucose, normalized to housekeeping gene abundance. Green bars: 2 h after switch from glucose medium to glucose medium, gray bars: non-/slow-growing cells 2 h after switch from glucose medium to fumarate medium, white bars: starved cells 2 h after switch from glucose medium to medium without a carbon source. Data from triplicate experiments. Error bars represent one standard error of the mean.

Source data are available online for this figure.

growing on glucose. Here, we found that ppGpp increases about 16-fold within the first 30 min after the shift (t-test P-value = 0.03) and then over the next 7.5 h decreases to about fivefold higher levels compared to those on glucose (t-test P-value = 0.03) (Fig 1C).

Another hallmark of stochastically induced persisters is the involvement of TAS genes (Maisonneuve et al, 2011). To determine whether the non-/slow-growing cells obtained after the nutrient shifts induce transcription of TAS genes, we performed reverse transcription and real-time PCR measurements of 13 different transcripts of the first genes in TAS operons. Here, we found that the shifts to fumarate and to medium without a carbon source both lead to increased abundance of TAS gene transcripts compared to cells growing exponentially on glucose (Fig 1D).

Thus, the nutrient-shift-induced persisters have the characteristics of the stochastically induced persisters: enhanced antibiotic tolerance, increased ppGpp levels, and increased expression of TAS genes. Remarkably, the slow-/non-growing cells in nutrient-rich conditions obtained after a nutrient shift have partly enhanced antibiotic tolerance to CCCP compared to starved cells, suggesting active tolerance mechanisms. In the following, we will call the nutrient-switch-induced persisters "persisters" and the starvation model persisters "starved cells".

**Persisters are metabolically active**

The enhanced antibiotic tolerance of persisters in nutrient-rich conditions could have been caused by energy availability, which, for instance, could be used to fuel multidrug-efflux systems. To test whether and how the nutrient-switch-induced persisters utilize nutrients (i.e. here, fumarate), we determined the physiological parameters of these cells, as well as starved cells and cells normally growing on fumarate. We found that in the first 2 h after the switch to fumarate or to medium without a carbon source, cells underwent a reductive division characterized by a decrease in cell volume (Fig 2A, Appendix Table S1) and an increase in cell count (Fig 2B). After that, the persisters grew at a rate of 0.02 ± 0.005 h⁻¹ (95% confidence interval) (Fig 2B) and their volume remained constant (Fig 2A, Appendix Table S1). As expected, the starved cells did not grow in number (Fig 2B), but they did in volume for the following 6 h (Fig 2A, Appendix Table S1). In other bacterial species, such an increase in volume was associated with swelling caused by a lack of membrane potential (Rottem et al, 1981). Thus, persister cells, in contrast to starved cells, must have metabolic activity to sustain their slow growth, and possibly to maintain their membrane potential.

Focusing on the nutrient and gas exchange rates in the persister cells, we found that they took up fumarate and oxygen and produced carbon dioxide, all at rates (per cell) approximately one order of magnitude lower than cells growing on fumarate (Fig 2C–E, Appendix Table S2). Looking at yields, compared to cells growing on fumarate, persisters produced 5.2 times less biomass per mol of consumed fumarate (Fig 3A), but exchanged more $O_2$ and $CO_2$ (2.1 and 1.3 times more per mol of consumed fumarate, respectively; Fig 3B). Thus, the differences between cells growing on fumarate and persisters extend beyond a simple scaling down of the respective metabolic rates. In fact, the higher $O_2$ and $CO_2$ yields, in combination with the lower biomass yield in persisters, indicate that persisters respire more and thus suggest that persisters produce more ATP per mol of consumed carbon source than fumarate-growing cells. We confirmed this conjecture with a flux balance analysis [using a genome-scale model of E. coli metabolism (Reed et al, 2003) and the measured physiological rates at 8 h (Appendix Table S2) with their 99.5% confidence intervals as constraints and assuming that they represent steady-state conditions as done before (Zampar et al, 2013)], which revealed that the maximum ATP yield (beyond what is needed for biomass production) in persisters is 8.4 times higher than in growing cells (Fig 3C). Thus, these data indicate that persisters operate their metabolism in a way that is optimized for energy generation, in contrast to biomass production and growth.

Through the flux balance analysis, we also found that persister cells, despite their very low fumarate uptake, could still produce ATP at a maximal rate of approximately 9.5 mmol gDW⁻¹ h⁻¹ (Fig 3D,

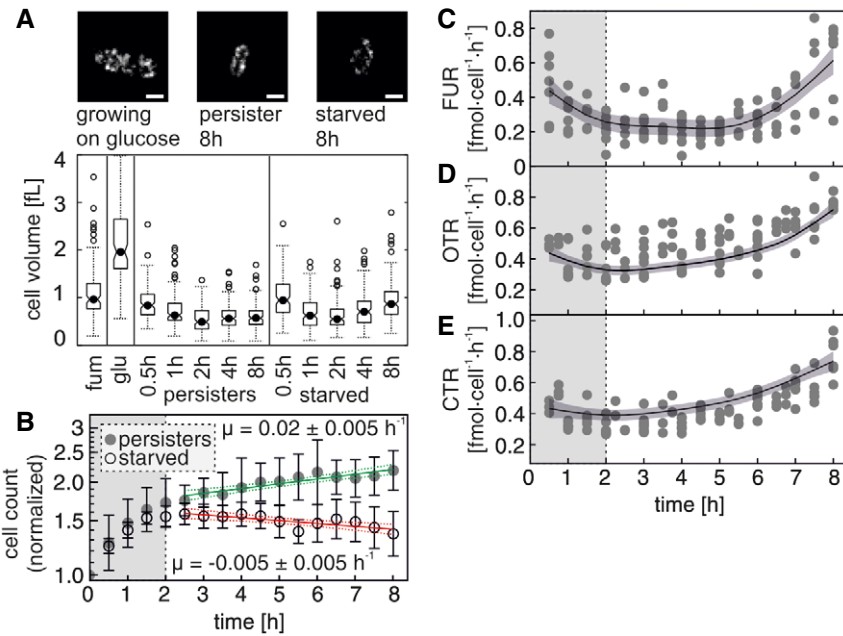

**Figure 2. Persisters grow and are metabolically active.**

A   Images of a cell growing on glucose, a persister cell and a starved cell. Scale bars: 1 μm. Volumes of cells growing on fumarate, cells growing on glucose, cells entering persistence and cells entering starvation. See also Appendix Table S1.

B   Evaporation-corrected cell count development of cell populations entering persistence and starvation. Gray disks: persister cells, open circles: starved cells. Values from each replicate were normalized to $t_0$. Error bars indicate one standard deviation. Green line (persister cells) and red line (starved cells) represent a prediction of a linear regression fitted to the log-transformed data, where slopes are equal to the growth rate with the dotted lines error margins representing the 95% confidence intervals determined by the model. Vertical gray area covering the period from 0 to 2 h visualizes the period of reductive division. Data from at least ten biological replicates.

C–E   Time course of the fumarate uptake rate (C), the oxygen transfer rate (OTR; D), and the carbon dioxide transfer rate (CTR; E) of persister cells. Points indicate time-specific rate values, and lines indicate fits from generalized additive models with 95% confidence interval (indicated by areas). Vertical gray area covering the period from 0 to 2 h visualizes the period of reductive division. Data from at least three biological replicates.

note that this is the ATP production rate beyond what is required for growth), a value which is enough to satisfy the non-growth-associated ATP maintenance requirements, which were estimated to between 8 and 10 mmol ATP $gDW^{-1}$ $h^{-1}$ (Orth *et al*, 2011). Through a metabolome analysis, we found that indeed the adenylate energy charge (AEC = ([ATP] + 0.5 [ADP])/([ATP] + [ADP] + [AMP])) was maintained at a high level in persisters (0.74, Fig 3E). The sum of adenylate nucleotide concentrations was identical in persisters and starved cells (Fig 3F), suggesting that the high adenylate energy charge is achieved thanks to energy generation (charging of AMP/ADP with phosphate). In comparison, cells growing on glucose and fumarate had an energy charge of 0.89 and 0.88, respectively (Fig 3E), with values higher than 0.8 being required for growth in *E. coli* (Chapman *et al*, 1971). In contrast, the adenylate energy charge in starved cells dropped to a value of 0.24 (Fig 3E), as expected from nutrient-deprived cells. Thus, through a metabolism focused on energy generation, persisters in nutrient-rich conditions are able to maintain high energy charges, eventually contributing to the observed enhanced antibiotic tolerance compared to starved cells.

**Persisters achieve a proteome state that starved cells fail to attain**

Toward determining the global proteomic phenotype of the persister cells, we next measured levels of about 2,000 proteins through liquid chromatography/mass spectrometry proteomics during entry into persistence from glucose (i.e. five time points until 8 h after the shift). For comparison, we measured protein levels in cells growing exponentially on fumarate and glucose, as well as during entry into starvation from glucose or fumarate. The data, that is, absolute numbers of protein copies per cell, are available in Table EV1.

Using the acquired data, we first aimed to identify the global differences between the proteome changes in persister and starved cells. Therefore, we used principal component analyses (PCA) to scale down the almost 2,000-dimensional dataset (across 14 different conditions/time points) into a two-dimensional PCA space (Fig 4, upper and middle panels). Here, we found that with time the persisters' proteome increasingly deviated from the proteome of glucose-grown cells (Fig 4A). Interestingly, although all cells were switched to fumarate, the proteome of persister cells did not approach the one of fumarate-growing cells. In fact, the proteomes of fumarate- and glucose-growing cells were more similar to each other than the proteome of persister cells compared to either the proteome of glucose-growing cells or fumarate-growing cells (Pearson's $r$ = 0.86, 0.75 and 0.66, respectively).

Furthermore, we found that the changes in the proteome of starved cells, which almost exclusively occurred during the first 2 h after the nutrient shift, followed the same trajectory as the proteome changes in the persister cells (Fig 4B). However, likely

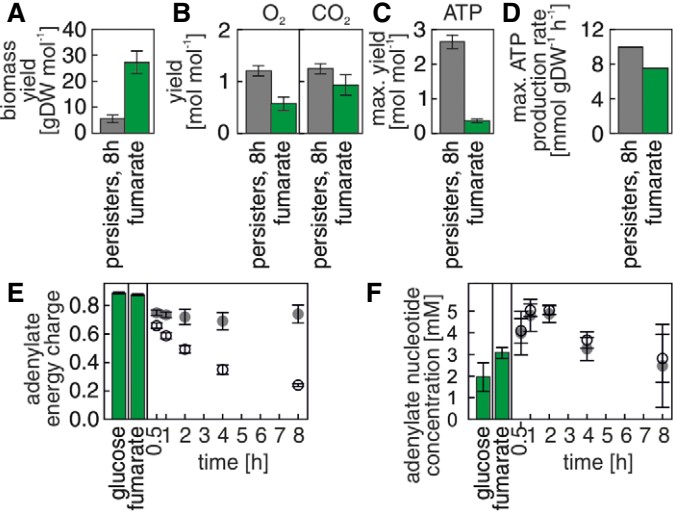

**Figure 3. Persisters maintain high energy charge levels through respiratory metabolism.**

A–C  Persister cells utilize a higher proportion of the taken up carbon for ATP production through respiration and less on biomass formation than cells growing on fumarate. The yields (relative to the up-taken fumarate) were calculated as ratios of physiological rates (cf. Materials and Methods) and in case of ATP, on the results of the flux balance analysis maximizing ATP production using the estimated physiological rates as constraints. Data from at least three biological replicates. Error bars indicate one standard error of the mean.

D  Maximal possible ATP production rates in persister cells and in fumarate-growing cells, estimated by flux balance analysis maximizing ATP production using the estimated physiological rates as constraints.

E, F  Adenylate energy charge (E) and sum of adenylate nucleotide concentrations (F). Gray disks: persister cells, open circles: starved cells, green bars: growing cells. Error bars indicate 95% confidence interval of the mean, calculated with a mixed effects model based on multiple biological replicates in multiple measurement campaigns (cf. Table EV2).

Source data are available online for this figure.

due to the lack of a carbon and energy source, the proteome of starved cells did not reach the same state as persister cells. Remarkably, the proteome of starved cells, which were previously grown on fumarate, also moved in the same direction as the proteomes of persisters and of starved cells generated from glucose-growing cells (Fig 4B). These findings implied that the proteome adjustments in both starved and persister cells must be caused by a common cue, which is not specific to the availability of a carbon source in the medium.

## Persister proteome is characterized by enhanced catabolism and $\sigma^S$-driven stress response

One factor inherent to the proteome changes of persister and starved cells is the downshift in growth rate (Fig 2B). Growth rate affects gene expression (Klumpp *et al*, 2009) and was identified to be a major factor in reorganizing proteome upon gradual glucose exhaustion (Berthoumieux *et al*, 2013). Thus, the observed proteome changes in the persister cells could be a simple consequence of the decrease in growth rate. To decipher whether the observed changes are a mere reflection of growth rate changes or whether the observed changes resemble specific characteristics of persisters, we made use of the proteome data that we recently generated when *E. coli* was grown on 11 different carbon sources and under three different stress conditions (Schmidt *et al*, 2016). We projected these proteomes onto the PCA space created by the proteomes of this study. Here, we found that the proteomes of cells growing on different carbon sources, and thus at different

growth rates, were mainly distributed along the dimension 2 of the PCA space, while the proteomes of the cells grown under the stress conditions moved along the dimension 1, as did the proteomes of persister cells and starved cells measured in this study (Fig 5A). The fact that persister cells and starved cells moved along the same direction as the stressed cells (i.e. pH, temperature, and osmotically stressed cells) suggests that a general stress response, or the stringent response triggered by the increased ppGpp levels (Fig 1C), is the driving force behind the specific proteome adjustments in persister and starved cells, instead of a global growth rate effect.

We next sought to identify the specific characteristics of the persister proteome. Therefore, we again used PCA together with a GOterm enrichment analysis, this time comparing persister cells at 8 h with the proteomes of growing cells, and separately with starved cells at 8 h (Fig 4, lower panel). In the first analysis, for which the results are shown in Fig 5B, we found that the proteins of "dimension 1" explain 75.2% of the total variation between the data of persister cells and growing cells. The enrichment analysis of this separating dimension showed that the persister proteome, compared to growing cells' proteomes, was characterized by lower levels of proteins for DNA replication, recombination, and SOS response (Fig 5B, see Appendix Table S3 for full ranked list). On the other hand, persister cells had higher levels of proteins responsible for stress response, including response to starvation, RNA catabolism, DNA repair, and protein folding (e.g. chaperones). In persisters, also a shift toward catabolism was observed (i.e. lower abundances of proteins for nucleotide, amino acid and cofactor

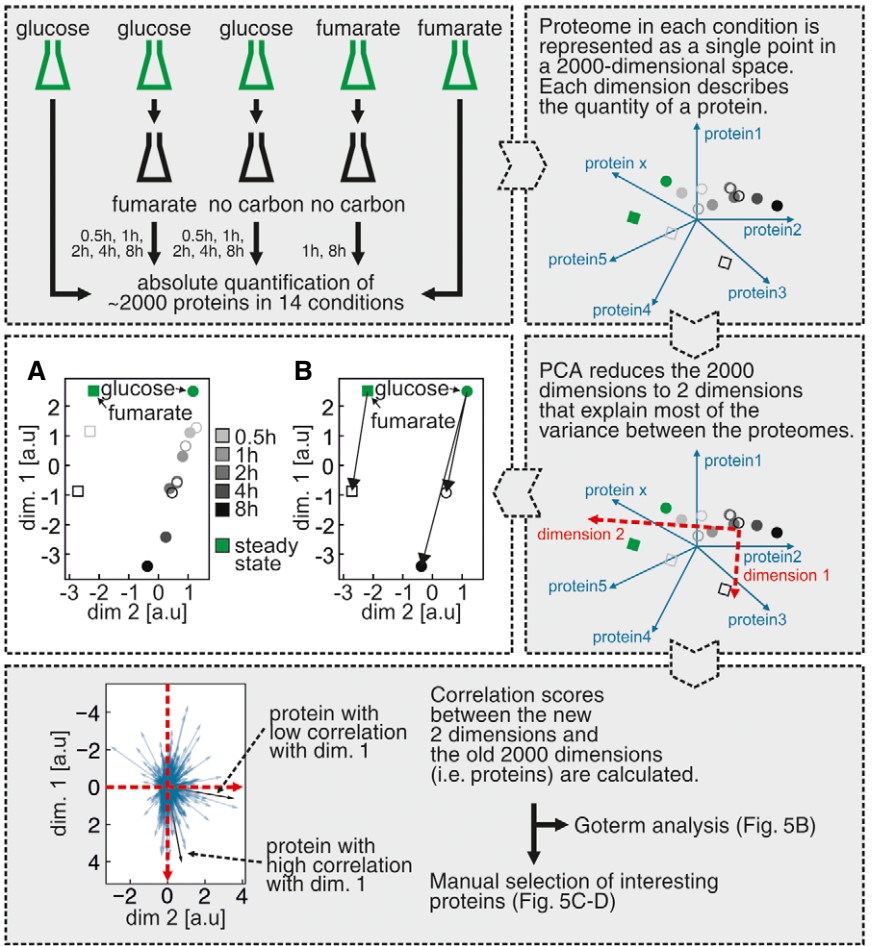

**Figure 4. Persisters' and starved cells' proteome are shaped by the same cue.**

The experimental and data analysis procedure is described in the gray boxes step by step.

A   PCA plot of the *Escherichia coli* proteomes in different conditions and time points. Each point represents a proteome in a different state. The distances between points are inversely correlated with the similarity between proteomes (i.e. proteomes with higher correlation coefficient have a shorter distance between each other), calculated based on differences in the expression level of each quantified protein. Green disk: cells growing on glucose, green square: cells growing on fumarate, gray disks: cells entering persistence after glucose-to-fumarate switch, open circles: cells entering starvation from glucose, open squares: cells entering starvation from fumarate. Time series are indicated by gray color gradients.

B   The progression of changes upon entry into starvation and entry into persistence happens in the same direction in the two-dimensional space, indicating that the same cue shapes these proteomes.

biosynthesis, and higher abundance of glycolysis proteins) (Fig 5B). When comparing persisters and starved cells, the analysis revealed similar processes to be upregulated as between persisters and growing cells (Fig 5B). This finding indicated—consistent with the analysis shown in Fig 4A—that the differences between the starved and the persister proteome lie mostly in the *strength* of protein expression changes and not in the *kinds* of proteins being expressed. Overall, the proteome of persisters (and to a lesser extent also the one of starved cells) was mostly characterized by a shift toward catabolism, as well as global stress response, compared to exponentially growing cells.

To identify proteins that particularly contribute to the observed phenotype of persister cells, we looked for proteins that were most significantly correlated with the separating "dimension 1" (which characterizes the persister-specific differences). One of these proteins (Fig 5C and D) is EmrA, a protein involved in CCCP export

and resistance to this drug (Lewis *et al*, 1994), which had a 1.6-fold higher concentration in persisters than in starved cells, eventually explaining the higher tolerance of persisters to this drug (cf. Fig 1B). Other notable differences included proteins involved in DNA replication, transcription, and translation. As for replication, we found higher levels of IHF (integration host factor) in persisters compared to both growing and starved cells (IhfA: 2.3 and 1.7-fold; IhfB: 2.4- and 1.6-fold, respectively). As the integration host factor has been found to enhance initiation of replication (Friedman, 1988), its higher abundance might point to an enhanced capability to initiate replication in persister cells, which would prime them for resuming growth. We also found higher levels of DnaC in persisters compared to cells growing on glucose (2.8-fold). DnaC was found to be crucial for restarting stalled replication forks (Nusslein-Crystalla *et al*, 1982) and required for initiation of replication (Kaguni *et al*, 1985). If indeed the higher levels of these proteins enhance the persisters

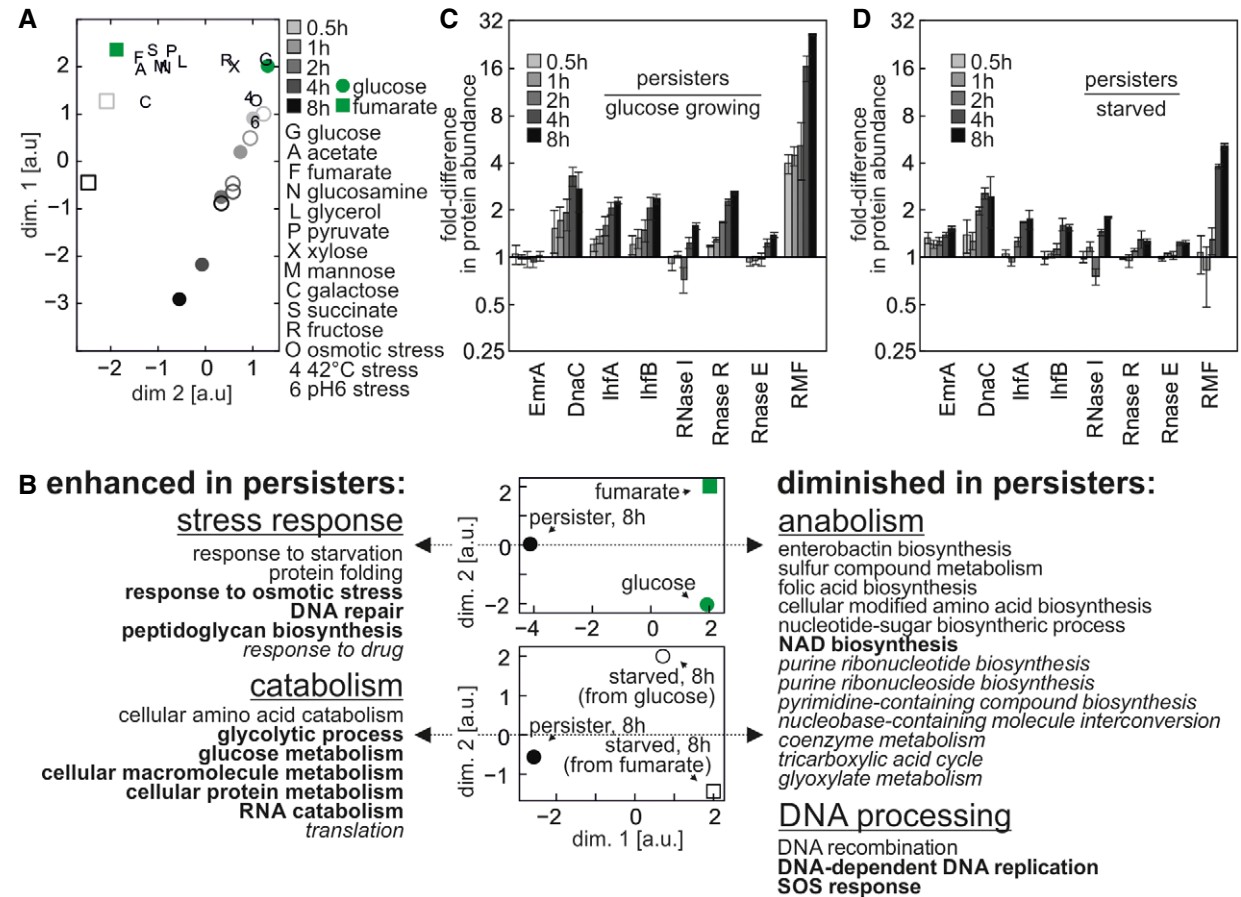

**Figure 5.  Proteome of persisters has enhanced catabolism and activation of stress response.**

A     Projection of *E. coli* proteomes in various growth and stress conditions (Schmidt *et al*, 2016) on the PCA space created by proteomes generated in this study.

B     PCA of proteomes of persister, fumarate-growing, and glucose-growing cells (upper panel); PCA of proteomes of persister cells and starved cells growing on glucose or fumarate before starvation (lower panel), markers as in (A). GOterms shared between the two analyses (i.e. persisters versus growing cells and persisters versus starved cells) are indicated in bold. GOterms specific to persisters versus starved cells analysis are indicated in italics. For a ranked list of assigned GOterms, see Appendix Table S3. See also Appendix Fig S2 showing expression levels of proteins involved in *Escherichia coli* central metabolic pathways.

C, D   Time profiles of abundance of selected proteins that are significantly correlated with the persister phenotype in both PCA (i.e. proteins for which the correlation coefficient had *P* < 0.1). Abundance relative to cells growing on glucose (C) or relative to starved cells (D). Error bars indicate one standard deviation reflecting variation between technical replicates.

Source data are available online for this figure.

cells' ability to replicate DNA, then potentially, persisters are ready to resume replication and growth.

As for translation, we found proteins involved in post-transcriptional regulation to be increased in persisters, compared to growing cells. Specifically, we observed higher levels of ribonucleases in persisters: RNase I (*rna*, 1.6-fold), R (*rnr*, 2.6-fold), and E (*rne*, 1.4-fold). While the two-first RNases are responsible for rRNA degradation (Kaplan & Apirion, 1974; Cheng & Deutscher, 2002), RNase R and RNase E are also responsible for mRNA degradation, the latter being also involved in specifically degrading transcripts essential for growth (i.e. ftsA-ftsZ) (Cam *et al*, 1996; Cheng & Deutscher, 2005). The persister proteome is also characterized by higher levels of RMF (ribosome modulation factor, 26.4-fold), which was found to cause the dimerization of the 70S ribosomes, also in persisters (Cho *et al*, 2015), leading to global inhibition of translation. These results suggest that in persisters, the cellular

processes of transcription and translation might be inhibited by mechanisms extending beyond toxin–antitoxin systems. The generally stronger overexpression of the above-mentioned proteins in persisters compared to starved cells again indicates that the main difference between the proteomes of persister cells and starved cells is caused by availability of nutrients and the rudimentary metabolic activity of persisters.

Next, we aimed to identify the regulatory factors responsible for the proteome of persisters. Here, $\sigma^S$ and the stress response was a good candidate, as the comparative analysis above suggested a stress response to be specific for the persister proteome. This notion was also in agreement with the fact that a lack of $\sigma^S$ affected the frequency of persister cell formation (Nguyen *et al*, 2011) and that $\sigma^S$ levels were increased in stochastically formed persister cells (Maisonneuve *et al*, 2013). Indeed, we found the levels of $\sigma^S$ to be increased upon entry into persistence (5.5-fold difference compared

to glucose-growing cells) as well as into starvation (3.9-fold difference compared to glucose-growing cells) and were maintained at about 1.5 times higher level in persister cells compared to starved cells 8 h after the medium switch (Appendix Table S4).

However, because mRNA of $\sigma^S$-induced genes could be subject to post-transcriptional regulation, just from the higher abundance of this sigma factor, we could not yet conclude that $\sigma^S$ is truly responsible for the shape of the persister proteome. However, when focusing on the genes regulated by $\sigma^S$ (Salgado *et al*, 2013), for which we had protein concentrations measured, we found that 67 out of 174 proteins were more than twofold overexpressed and only 11 out of 174 were more than twofold depleted in persisters compared to glucose-growing cells. In starved cells, we found 41 out of 174 proteins to be more than twofold overexpressed and 11 out of 174 to be more than twofold depleted compared to glucose-growing cells (Appendix Table S5).

To establish that the changes in persister proteome were indeed governed by $\sigma^S$, we performed a hypergeometric test using known sigma factor–gene interactions, transcription factor (TF)-gene interactions, and regulatory RNA-gene interactions (all from the RegulonDB database; Salgado *et al*, 2013) on genes with at least twofold change in protein abundance, selected by clustering along similar profiles of protein concentration change in time with STEM software. Here, we found that in persister cells, $\sigma^S$ had the lowest *P*-value of all regulatory factors (FDR-adjusted *P* = 0.015; for all other sigma factors and TF factors as well as regulatory RNA the *P*-values were higher than 0.05). This finding indicated that the proteome changes in persister cells were largely controlled by $\sigma^S$.

Consistent with the general function of stress response to limit the demand for resources and to ensure survival (Hardiman *et al*, 2007; Shimizu, 2013), our PCA did not reveal any changes in the persisters' metabolic proteome toward adapting these cells to the new condition (i.e. growth on fumarate in this case). The revealed upregulation of glycolytic proteins (less than twofold compared to glucose-growing cells, Fig 5B, Appendix Fig S2) is not needed for growth on fumarate, as it is a gluconeogenic substrate. In fact, the only major differences we found in central metabolic enzymes between persister and growing cells (i.e. the fumarate reductases and the aconitase) can be explained by the regulatory action of $\sigma^S$.

Overall, we found that $\sigma^S$ is primarily responsible for the proteome of persisters. Because we also found that $\sigma^S$ is upregulated in starved cells (Appendix Table S4), the cue for entering the persister state in nutrient-rich conditions must be the same as for entering starvation. As $\sigma^S$ was upregulated for the whole period of our observation in persisters and starved cells (Appendix Table S4), this suggests that the cue inducing the $\sigma^S$ stress response must be sustained over time. Because persisters maintain metabolic activity, change protein expression, and grow, it is interesting to ask what is the cue, or the factor, which causes them to perceive starvation even though nutrients are available. Increased ppGpp levels cause an upregulation of $\sigma^S$ expression (Gentry *et al*, 1993). As we found elevated ppGpp levels in persisters (Fig 1C), these increased ppGpp concentrations could be the cause for the higher abundance of $\sigma^S$ in persister cells. However, it is still a question which mechanism would cause ppGpp levels to be increased in persisters, and what is the cue triggering this mechanism.

## Persister cells and starved cells share a distinct metabolite pool pattern

While the proteome of both persisters and starved cells was triggered in response to likely the same cue, our physiological characterization demonstrated that this signal cannot be the cells' energetic state, or a lack of metabolic activity, because these two features were strongly different in persisters and starved cells. However, intracellular metabolite concentrations, which can regulate gene expression (cf. Kochanowski *et al*, 2013), could serve as the signal triggering the stress response in persister cells present in nutrient-rich conditions as well as in starved cells. In search for such a metabolic cue, we measured the concentrations of 29 metabolites of central carbon metabolism as well as of energy and redox cofactors at different time points during entry into persistence and starvation, and in cells growing exponentially on fumarate and on glucose.

Here, we found that compared to growth on glucose and fumarate, the concentrations of most glycolytic and pentose phosphate pathway intermediates dropped one to two orders of magnitude to concentrations in the μM range in the first 30 min of entry into persistence and starvation (Fig 6, Table EV2). The

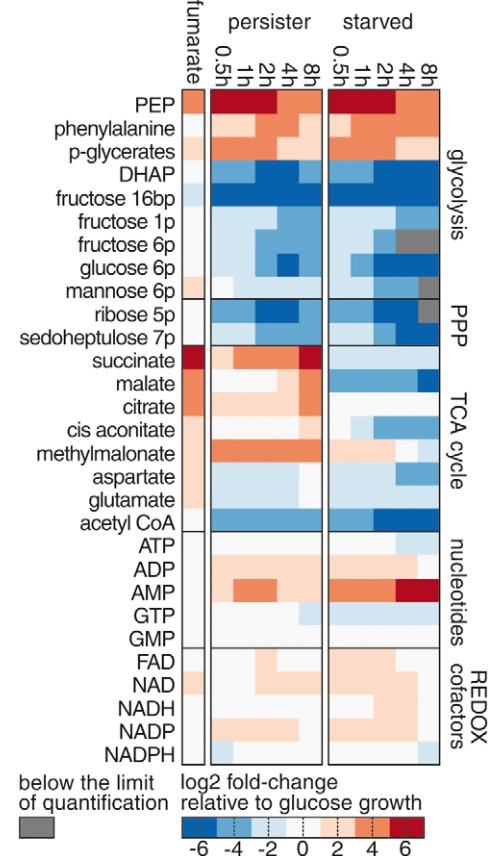

**Figure 6.  Persister and starved cells have depleted metabolite pools.**
Change in metabolite concentrations in persister, starved, and fumarate-growing cells relative to glucose-growing cells. For absolute concentrations with error estimates and numbers of replicates, see Table EV2.

only glycolytic metabolites with higher concentrations in persisters and starved cells, compared to glucose-grown cells, were phosphoenolpyruvate (PEP) and the PEP derivatives, phosphoglycerates, and phenylalanine. The tricarboxylic acid cycle intermediates, most probably due to fumarate as the chosen carbon source, had higher concentrations in persisters compared to the starved cells, as high as in fumarate-growing cells. The distinct concentration pattern of glycolytic metabolites (i.e. extremely low levels with the exception of increased levels of PEP and phosphoglycerates), similar in starved cells and persister cells, could contain the signal for the entry and sustenance of the stress response.

## Low metabolic flux causes persister formation without σ^S or TAS action

On the basis of our generated data and previous persister cell studies, we developed a conceptual model for the entry into and the sustenance of the persister state (Fig 7A). At the core of this model is a positive feedback loop that can drive cells into and arrest them in the persister state. According to this model, a trigger for entry into persistence is strong perturbations of metabolic homeostasis—perturbations beyond the intrinsic buffering capacities of enzymes or beyond possible adjustments of enzyme levels—leading to critically lowered metabolic fluxes. At such low fluxes, a correction of

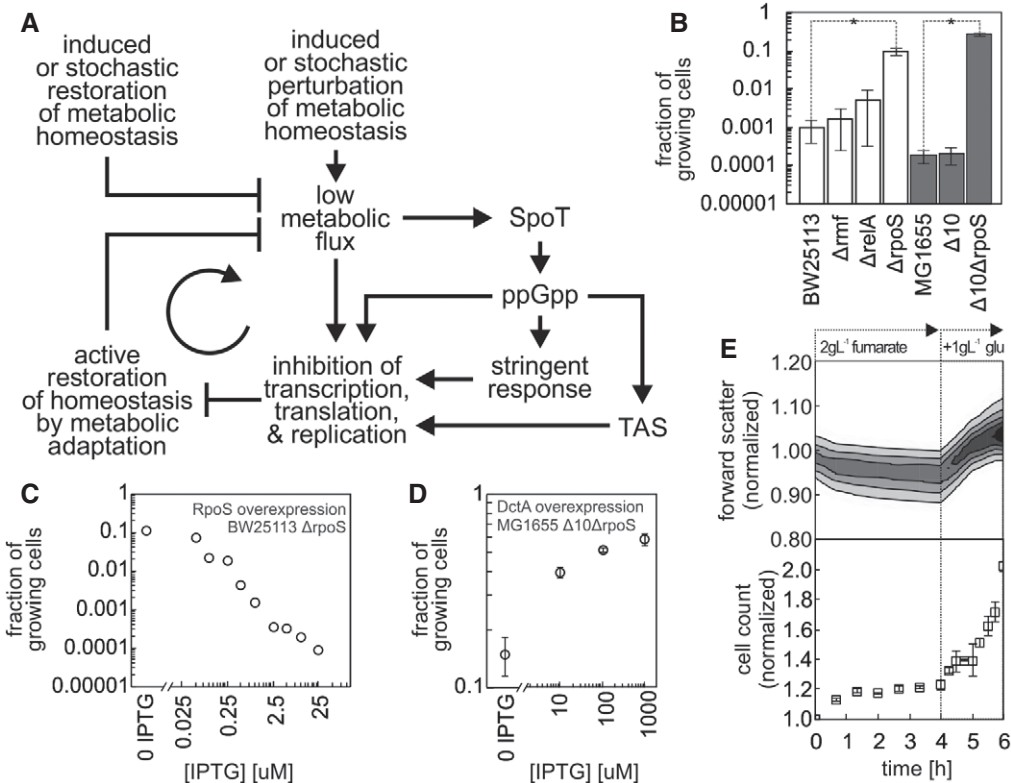

**Figure 7. Persistence is sustained through a system-level feedback loop.**

A   A metabolic perturbation beyond the intrinsic buffering capacity of metabolism, which results in low metabolic flux, is the trigger for persistence. Cells with critically low metabolic fluxes get into a vicious cycle (feedback) and thus cannot restore metabolic homeostasis. The robustness of this primitive, system-level feedback loop can be enhanced via various mechanisms, such as action of TAS, σ^S, or ppGpp, which lead to further inhibition of transcription and translation. The system-level feedback loop is active until the vicious cycle is broken through restoration of metabolic homeostasis, for example, by addition of certain nutrients, stochastically higher expression of certain flux-limiting enzymes or stochastically low expression of growth-inhibiting mechanisms.

B   Fraction of growing cells (i.e. 1 − fraction of persister cells) in various knockout strains after a glucose-to-fumarate nutrient shift. White bars: BW25113-derived strains. Gray bars: MG1655-derived strains. Δ10: strain with 10 TAS knockout. Statistically significant difference (*t*-test or Wilcoxon rank sum test for Δ*rmf*, *P*-value < 0.05) between the mutants and the respective wild type is indicated with an asterisk (*). Mean of at least three replicates and standard deviations are shown. See Appendix Table S6 for antibiotic tolerance assay results.

C   Fraction of growing cells after a glucose-to-fumarate shift decreases with higher induction of σ^S expression in Δ*rpoS* strain. Cells were induced with IPTG at the indicated concentrations after the nutrient shift.

D   Fraction of growing cells after a glucose-to-fumarate shift increases with higher induction of DctA fumarate transporter, and thus with higher metabolic flux, in the Δ10Δ*rpoS* strain. Cells were induced with the indicated IPTG concentrations for 16 h prior to the nutrient shift and after the nutrient shift. Mean of three replicates and standard deviations are shown.

E   Rapid increase in forward scatter and cell count upon addition of glucose (at 4 h after nutrient shift) indicates that persister cells can rapidly resume growth upon externally driven restoration of metabolic homeostasis. Forward scatter distribution from three replicates, cell count mean of three replicates, and standard deviations are shown.

metabolism is impossible, because rates of protein synthesis could be lower than the rates of protein degradation. As such, at critically low metabolic fluxes, it might be impossible for cells to restart metabolism or to re-adjust metabolic homeostasis. Together, these mechanisms could form a sort of vicious cycle (i.e. feedback) leading to persistence (Fig 7A). It should be noted that this feedback mechanism does not involve toxins and is therefore different to the one proposed by (Klumpp et al, 2009), who suggested a toxin-based feedback mechanism resulting in growth bistability during steady-state growth.

Our model further suggests that the robustness of such "low-level" vicious cycle could be enhanced by the stress response and toxin/antitoxin systems, as well as other inhibitory mechanisms such as RMF-induced ribosome dimerization. These mechanisms could strengthen the feedback by inhibiting translation, transcription, and replication, or by allocating the limiting resources (under the perturbation of metabolic homeostasis) to stress response. Such allocation of resources would further reduce (or prevent) metabolic adjustments or corrections, although with the resources still available it might also be possible to attempt restoring metabolic homeostasis. These feedback-enhancing mechanisms would likely ensure bacterial survival during strong perturbations of metabolic homeostasis, by prohibiting that cells engage in "risky" metabolic adjustments and instead invest in stress response and maintenance given the available limited resources (indicated by low metabolic fluxes).

On the basis of the identified characteristic metabolite pool pattern, we hypothesize that activation of the observed stress response could occur through a metabolite, which modulates the activity of (or being one of the substrates for) the ppGpp synthases

RelA or SpoT leading to increased levels of ppGpp to trigger the stringent and other stress responses. According to our model, perturbations in components such as the TAS, $\sigma^S$, or ppGpp synthesis should either sensitize or de-sensitize the feedback loop depending on the nature of the perturbation, but not abolish the feedback loop between low metabolic fluxes and lack of metabolic adjustment/correction. Thus, TAS, $\sigma^S$, or ppGpp synthase perturbations should only modulate the fraction of persister cells, but not eliminate them.

To test whether the currently known growth-inhibiting persistence mechanisms, TAS and RMF, indeed only modulate the fraction of persister cells, we switched various mutant strains from glucose to fumarate and assessed the number of persisters emerging: We tested $\Delta rmf$ (ribosome modulation factor, inhibitor of protein synthesis) in our wild-type background (BW25113), as well as the 10 TAS knockout strain generated in Kenn Gerdes' laboratory ($\Delta$10, Maisonneuve et al, 2011) and the respective parental strain, MG1655. Here, we found that deleting neither the TAS systems nor the ribosome modulation factor changed the fraction of cells adapting to fumarate (Fig 7B), and thus, the fractions of persisters formed. Next, we investigated the role of $\sigma^S$ and stress response, which, on the basis of our proteome analyses, we speculated to have a significant role in establishing the persister phenotype. First, consistent with the fact that the ppGpp synthase RelA is primarily involved in amino acid starvation response (Haseltine & Block, 1973), we found that its deletion did not modulate the fraction of adapting cells and thus not the amount of persisters formed (Fig 7B). This finding suggests that here rather SpoT (the second E. coli ppGpp synthase) might be responsible for synthesizing ppGpp and for its increased

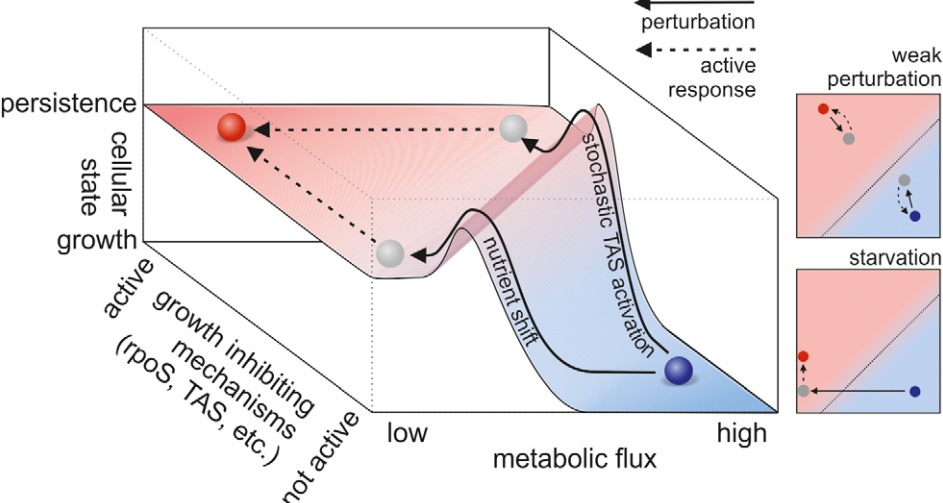

**Figure 8.   Schematic model: Persistence and growth are two attractor states on a phenotypic landscape with the dimensions "metabolic flux" and "activity of growth-inhibiting mechanisms".**
The blue circle denotes the normal growth state, the red circle denotes the persister state, and the gray disk indicates the position of a cell directly after a perturbation. The magnitude of metabolic flux and the activity of growth-inhibiting mechanisms determine a cell's position on the landscape. If a cell is on the right side of the watershed (i.e. the hill/dotted line), it will move toward the attractor indicated by the blue disk and achieve normal growth in metabolic homeostasis. If a cell happens to be on the left side of the watershed, it will become a persister cell. Both states are achieved through active mechanisms (that eventually also require resources/energy), as indicated by the finding that the persister state is not equal to the starved state. Various perturbations that were found to cause persistence (for instance, stochastic TAS induction, nutrient shift, or diauxie) move the cell on the landscape in different directions, but all of them push it from the state of metabolic homeostasis beyond the watershed.

concentration (Fig 1C). As a deletion of *spoT* was never achieved without obtaining spontaneous suppressor mutations in the *relA* gene (Montero *et al*, 2014), and the double knockout strain Δ*relA*Δ*spoT* cannot grow without certain amino acids (Xiao *et al*, 1991), we directly tested the Δ*rpoS* strain (*rpoS* encodes for σ$^S$). Deleting *rpoS* in the BW25113 background and in the Δ10 strain both significantly increased the number of cells adapting to fumarate (*t*-test *P*-value = 0.0025 and 0.0032, respectively) (Fig 7B) and thus decreased the number of persisters. To perturb the system in the opposite direction, we complemented the BW25113 Δ*rpoS* strain with a plasmid for IPTG-inducible expression of σ$^S$ and switched the cells from glucose-to-fumarate medium supplemented with different IPTG concentrations. Here, we found that with progressively higher concentrations of IPTG (and thus higher σ$^S$ expression levels), more cells assumed the persister phenotype (Fig 7C). These findings show that σ$^S$ modulates the strength of the feedback and thus the amount of persisters, most probably as a response to SpoT activity. However, persisters also occurred in absence of σ$^S$ (Fig 7B). Therefore, also σ$^S$ is not essential, but still plays a role in establishing the persister state.

Because the growth-inhibiting mechanisms currently thought to be responsible for persister formation did not lead to complete elimination of persister cells, these findings provide further evidence toward the proposed metabolic flux-dependent primitive vicious cycle forcing cells into persistence (Fig 7A). Critical perturbations of metabolic homeostasis leading to lowered metabolic fluxes could be low expression (for instance, for stochastic reasons) of flux-controlling enzymes or nutrient transporters (Kiviet *et al*, 2014), drastic nutrient shifts, reductions of nutrient influx, or complete nutrient deprivation (for instance, during stationary phase). In fact, it has been shown that the fraction of persisters inversely correlated with glucose influx (Maisonneuve *et al*, 2013), and with acetate or fumarate influx (Kotte *et al*, 2014). To test whether the entry into persistence is still metabolic flux dependent in a strain that lacks TAS and σ$^S$, which would provide further support for the primitive vicious cycle, we transformed the Δ10 Δ*rpoS* strain with a plasmid for IPTG-inducible expression of the fumarate transporter DctA, through which we could previously modulate the metabolic flux upon shifts to fumarate (Kotte *et al*, 2014). Here, we found that with increasing DctA expression and thus increasing fumarate uptake flux, the fraction of persisters indeed decreased (Fig 7D). This finding, together with the fact that the Δ10 TAS Δ*rpoS* strain still produced persisters upon a nutrient shift (Fig 7B), provided further support to our model, in which the metabolic flux is the basic factor in establishing persistence, while other mechanisms enhance the feedback.

Finally, according to our model, a change to beneficial environmental conditions should immediately break the vicious cycle by enabling persister cells to regain homeostasis passively without adjustment of the metabolic machinery. To test whether this is indeed the case, we added glucose to the persister cells 4 h after the shift to fumarate. We found that the cells indeed started growing in size (measured via forward scatter) and in number, virtually immediately after the addition of glucose (Fig 7E). This finding suggests that the factors inhibiting persister growth can be removed on a very short timescale.

## Discussion

Using a recently proposed way to generate persisters in large quantities, and high-throughput analytical methods, we comprehensively mapped the molecular phenotype of cells during the entry and in the state of persistence in nutrient-rich conditions. We found that *E. coli* persisters in nutrient-rich conditions take up nutrients and grow slowly through a metabolism that is focused on energy production, although these cells could have utilized the available nutrient to adapt to the new conditions and ultimately grow faster. Still, their rudimentary metabolism accompanied by depleted metabolite pools (Fig 6) is sufficient to generate enough ATP to sustain non-growth-associated maintenance costs (Fig 3D) and a high adenylate energy charge (Fig 3E). The proteome of *E. coli* persisters, which the starved cells try to, but do not, reach (Fig 4A) possibly due to lack of energy or carbon, does not show any signs of metabolic adaptation. Instead, the persister proteome is characterized by shift toward catabolism and stress response caused by the action of σ$^S$ (Fig 5B). Likely due to the σ$^S$-shaped proteome and the sustained energy charge (Fig 3E), persisters in nutrient-rich conditions exhibit higher tolerance to certain antibiotics in comparison with starved cells (Fig 1B). On the basis of the generated data and the previous findings in the field, we developed a conceptual system-level model on the emergence of and sustenance in persistence.

According to this model, the state of persistence and the state of normal growth can be thought of as two attractors on a phenotypic landscape divided by a watershed (Fig 8). Moving on this landscape can, on the one hand, be accomplished by changing the magnitude of metabolic fluxes and on the other hand by altering the activity of growth-inhibiting mechanisms. Cells in metabolic homeostasis have high flux and low activity of growth-inhibiting mechanisms. Once the watershed is crossed, for instance through a critically lowered metabolic flux (i.e. caused by a nutrient shift) or a critically increased activity of growth-inhibiting mechanisms (i.e. through stochastic induction of TAS activity), cells will be attracted to the persister state. Achieving the ultimate persister state is an active process, which requires metabolic activity, active protein expression and thus presence of a carbon source. A perturbation that does not move the phenotypic state of the cell beyond the watershed will allow cells to restore metabolic homeostasis and continue to grow, not assuming the persister state.

Our model for persistence integrates the different (and at the first view contradicting) findings on persisters: While at the model's core is a flux-dependent feedback, there is a multitude of molecular mechanisms that can modulate the feedback. Because these two factors (metabolic flux and growth-inhibiting molecular mechanisms) can affect persister frequency, it is of crucial importance that they are both considered when investigating bacterial persistence, as also pointed out recently (Kaldalu *et al*, 2016). We envision that our developed model, which unifies the current and the newly obtained knowledge about persistence and creates a system-level view on persistence, will provide an important basis for future research toward understanding and eradicating persisters.

# Materials and Methods

## Bacterial strains used in this study

| Strain | Source |
|---|---|
| BW25113 | Obtained from (Baba *et al*, 2006) |
| BW25113 Δ*rmf* | Obtained from (Baba *et al*, 2006) |
| BW25113 Δ*relA* | Obtained from (Baba *et al*, 2006) |
| BW25113 Δ*rpoS* | Obtained from (Baba *et al*, 2006) |
| BW25113 Δ*rpoS* + pNT3-rpoS | BW25113 Δ*rpoS* transformed with pNT3-rpoS plasmid from (Saka *et al*, 2005) |
| BW25113 + pBAD-LacY-EYFP | BW25113 transformed with pBAD-LacY-EYFP plasmid (gift from Jonas van der Berg) |
| MG1655 | Obtained from (Maisonneuve *et al*, 2011) |
| MG1655 Δ10 | Obtained from (Maisonneuve *et al*, 2011) |
| MG1655 Δ10Δ*rpoS* | MG1655 Δ10 with rpoS knockout transduced with P1 phage from BW25113 Δ*rpoS* |
| MG1655 Δ10Δ*rpoS* + pNT3-dctA | MG1655 Δ10Δ*rpoS* transformed with pNTR-SD-dctA plasmid from (Saka *et al*, 2005) |

## Media and cultivation

*Escherichia coli* K12 strain BW25113 was used for the phenotypic characterization. Parts of the model validation were also done with the strain MG1655. All experiments were performed using M9 minimal medium, which was prepared as previously described (Kotte *et al*, 2014) or LB that was autoclaved and then filtered through a 0.2 μm polyethersulfone (PES) filter. M9 medium was supplemented with a carbon source to a final concentration of 5 g l$^{-1}$ glucose or 2 g l$^{-1}$ fumarate, unless indicated otherwise. The carbon source stock solutions were made by dissolving the carbon source in demineralized water, adjusting the pH to 7 with NaOH or HCl, and filtering through a 0.2 μm PES filter. Cultivations were done in 50 ml of M9 medium in a 500-ml Erlenmeyer flask closed with a 38 mm silicone sponge closure (Bellco Glass) at 37°C, 300 rpm, and 5 cm shaking diameter. On the following day, cells were diluted into a new culture prepared in the same way as the overnight culture, incubated, and further diluted as needed in order to keep the cells in mid-exponential phase.

## Flow cytometric analyses

Cell counts, fluorescence intensity, and forward scatter values were determined with an Accuri C6 flow cytometer. Cells were diluted to an appropriate density with M9 medium without a carbon source directly prior to analysis. The flow cytometer was set to measure 20 μl volume, with the fluidics setting set to "medium". The SSC-H and FSC-H thresholds were set to 500 and 8,000, respectively, in order to cut off most of the electronic noise. The Accuri CFlow Plus software was used for data analysis.

## Antibiotic tolerance test

Cell were switched from M9 medium with glucose to M9 medium with fumarate with (for time course experiments with ampicillin) or

without staining (for all other experiments), as described previously (Kotte *et al*, 2014). Staining was performed using the PKH-67 dye (Sigma-Aldrich). At various time points, antibiotics were added and the cells were incubated for 2 h at 37°C with shaking. Thereafter, 0.5 ml of the culture was transferred to 50 ml pre-warmed LB medium. Cells that resumed growth became bigger, and non-growing cells retained their size. The fraction of cells that do not resume growth was determined by observing cell size changes (with flow cytometry via forward scatter, a value which correlates with cell size) under the new conditions every 30 min from 0 to 4 h after the transfer to LB medium. Then, fractions of non-recovering cells from the period between 2 and 3 h were averaged for each replicate. A control experiment without antibiotics was performed, in order to measure the machine noise and the eventual fraction of non-recovering cells that died not due to antibiotic action, but due to the medium switch procedure. This level of non-recovering cells and noise was used to normalize the results.

## Relative quantification of TAS transcripts abundance

To determine the relative amount of TAS transcripts, $0.9 \times 10^9$ cells growing on M9 medium supplemented with 5 g l$^{-1}$ glucose, $1.6 \times 10^9$ cells switched to M9 medium without carbon source or supplemented with 2 g l$^{-1}$ fumarate, and $0.9 \times 10^9$ cells growing on M9 medium supplemented with 5 g l$^{-1}$ glucose that were treated in an identical way as during a nutrient switch were mixed in a 4:1 ratio with a 5% solution of phenol in ethanol (ice-cold) and incubated for 30 min on ice in order to stop RNA metabolism. Bacteria were then centrifuged (5 min, 8,000 *g*, 4°C), resuspended in the same volume of 1% solution of phenol in ethanol (ice-cold), centrifuged, (5 min, 8,000 *g*, 4°C), resuspended in the same volume of PBS (ice-cold), and centrifuged again (5 min, 8,000 *g*, 4°C). Pellets were then lysed with 150 μl of 1 mg ml$^{-1}$ lysozyme solution in TE buffer (10 mM Tris–HCl pH 8.0, 1 mM EDTA) for 5 min at RT. Then, total RNA was extracted with 1 ml of TRIzol (Invitrogen), followed by two extractions with 0.5 ml of chloroform. RNA was precipitated in 1 ml of 0.3 M sodium acetate 70% ethanol solution at −20°C for at least 16 h, pelleted by centrifugation (10 min, 12,000 *g*, 4°C), and washed twice with 1 ml of 75% EtOH. RNA was then dissolved in 20 μl of RNase-free water and treated with 3 U of DNase I for 1 h at 37°C (Turbo DNA-free, Ambion/Applied Biosystems). After DNase I inactivation, RNA integrity was assessed by agarose–TAE electrophoresis, and the yield and purity of the RNA was determined by spectrophotometry (Nanodrop Instruments). Genomic DNA contamination of RNA samples was ruled out by performing a 30-cycle PCR using *gapA* primer pair and 150 ng of RNA of each sample and verifying that no product was synthesized. For cDNA library construction, 1 μg of total RNA was reverse-transcribed with random hexamers using the High-Capacity cDNA Archive kit (Applied Biosystems) in a one-step run of 10 min at 25°C, 2 h at 37°C, and 5 min at 85°C. For real-time quantitative PCR (qPCR) analysis, 2 ng of the cDNA library as template, 0.5 μM of each primer, and the Power SYBR Green PCR master mix (Applied Biosystems/Life Technologies) in a 10 μl final volume were used. qPCRs were performed in an ABI Prism 7300 instrument (Applied Biosystems) using the following program: 10 min at 95°C; 45 cycles of 15 s at 95°C and 1 min at 60°C; dissociation curve of 15 s at 95°C, 1 min at 60°C; and a progressive temperature increase until

95°C. All qPCR runs included a non-template control, a mock cDNA library produced without reverse transcriptase, and a standard curve with different amounts cDNA. When feasible, primers were designed using Primer Express 3.0 (Applied Biosystems), and the quality of manually designed primers was examined with the same software. Primers are listed in Appendix Table S6. For data analysis, the amplification efficiency in each qPCR assay was determined using a standard curve with 20, 4, 0.8, 0.16, 0.032, and 0 ng of cDNA run in parallel with the test samples. The baseline and threshold were set using the standard curve, and only cycle threshold ($C_t$) data from the samples lying within the linear range of the standard curve were considered. The $C_t$ for every transcript in each sample corresponds to the mean value of the Cts obtained in three qPCRs run in parallel. $C_t$ data from each sample were subtracted from the mean $C_t$ value obtained from bacteria growing in M9 supplemented with glucose before the switch ($\Delta C_t$). These values were converted into relative fold expression by raising the amplification efficiency of each qPCR assay to the $-\Delta C_t$ value. Expression data were further normalized to the mean relative expression value of six housekeeping genes (Appendix Table S7). Relative expression was assessed in three independent biological replicates for each experimental condition.

### Determination of fumarate uptake rate of persister cells

To determine the fumarate uptake rate (FUR) of persister cells, exponentially growing *E. coli* cultures on M9 medium supplemented with glucose were harvested, centrifuged (10 min, 4,000 g, 4°C), washed twice with M9 medium containing no carbon source, and resuspended in M9 2 g l$^{-1}$ fumarate medium resulting in a cell concentration from $1 \times 10^9$ ml$^{-1}$ to $5 \times 10^9$ ml$^{-1}$. The new cultures were incubated, and samples were taken over a period of 8 h. Cell counts in the samples were determined by flow cytometry. For each sample, the medium was centrifuged (5 min, 4,000 g), filtered through a Spin-X centrifuge filter (Corning, 0.22-μm nylon membrane, 3 s, 4,000 g), and stored at 4°C. Fumarate concentration was determined by HPLC using a Hi-Plex H ion exchange column (MP: 0.005 M H$_2$SO$_4$, 0.6 ml min$^{-1}$, isocratic) and detection of UV absorbance at 210 nm. The fumarate concentration was then determined with a calibration curve prepared freshly for each experiment by dissolving appropriate amounts of sodium fumarate in M9 medium. The time-specific uptake rate was calculated as follows:

$$r_{t_i} = \frac{S_{t_{i-1}} - S_{t_i}}{(t_{i-1} - t_i)X_{t_i}},$$

where $S$ is the fumarate concentration, $t$ is the time elapsed from the beginning of the experiment, and $X$ is the cell count. Then, a generalized additive model using univariate penalized cubic regression spline smooths with eight knots was fitted to the data using R in order to estimate the mean and error values.

### Determination of oxygen and carbon dioxide transfer rates of persister cells

*Escherichia coli* cultures were prepared as for fumarate uptake rate determination. All experiments were performed in technical triplicates using the Respiration Activity Monitoring System (RAMOS) in special flasks filled with 26 ml of medium, shaken at 300 rpm. Cell counts and fumarate uptake rate were determined in parallel in a separate flask at an identical cell density. Ten minutes of rinsing time and 50 min of measurement time in the RAMOS device were used. Then, a generalized additive model was fit to the data using R in order to estimate the mean and error values as for fumarate uptake rate estimation.

### Determination of fumarate uptake rate and oxygen and carbon dioxide transfer rates in growing cells

To determine the fumarate uptake rate of cells exponentially growing on fumarate, a culture containing fumarate-adapted cells was diluted to a cell concentration that allowed for about 20 h of growth before fumarate depletion. About 8 h prior to fumarate depletion, samples were taken every 30 min. Cell counts in the cultures were determined by flow cytometry. Samples for determination of fumarate concentration were taken and analyzed with HPLC-UV as for persister cells. In parallel to the culture made for fumarate uptake rate determination, three cultures (technical triplicates) were made to measure oxygen and carbon dioxide transfer rates. All experiments were performed using the Respiration Activity Monitoring System (RAMOS) in special flasks filled with 26 ml of medium, shaken at 300 rpm. Ten minutes of rinsing time and 20 min of measurement time were used. All calculations were performed by the software package provided with the RAMOS system, which returned time-specific total carbon dioxide and total oxygen transfer values. The fumarate concentration-dependent fumarate uptake rate, fumarate concentration-dependent oxygen transfer rate, fumarate concentration-dependent carbon dioxide transfer rate, and fumarate concentration-dependent growth rate were determined by fitting the cell counts and fumarate concentration measurements, as well as the cumulative oxygen transfer and carbon dioxide transfer values to a model describing exponential growth of cells with a fumarate concentration-dependent growth rate following Monod kinetics:

$$\frac{dX}{dt} = \mu X,$$

$$\frac{dS}{dt} = \frac{1}{Y_{XS}}\mu X,$$

$$\frac{dO}{dt} = \frac{1}{Y_{OS}}\mu X,$$

$$\frac{dC}{dt} = \frac{1}{Y_{CS}}\mu X,$$

$$\mu = \frac{\mu_{max}S}{K_S + S},$$

where μ is the growth rate, S the fumarate concentration, $X$ the cell count, O cumulative oxygen transfer, C cumulative CO$_2$ transfer, $\mu_{max}$ the maximum growth rate, $K_S$ the Monod constant, $Y_{XS}$ the cell count/fumarate yield, $Y_{OS}$ cell count/oxygen yield, and $Y_{XS}$ the cell count/carbon dioxide yield. The fitting was performed in MATLAB, using MCMC toolkit (Haario *et al*, 2006). The fumarate uptake rate, oxygen transfer rate, and carbon dioxide transfer rate of growing

cells were then determined with the estimated parameter values (± SD):

$$\mu_{\max} = 0.99 \pm 0.05 [\text{h}^{-1}]$$

$$K_S = 1.53 \pm 0.18 [\text{g l}^{-1}]$$

$$Y_{XS} = 0.90 \times 10^{12} \pm 0.15 \times 10^{12} [\text{g}^{-1}]$$

$$Y_{OS} = 1.83 \times 10^{10} \pm 0.58 \times 10^{10} [\text{mol}^{-1}]$$

$$Y_{CS} = 1.11 \times 10^{10} \pm 0.31 [\text{mol}^{-1}]$$

and the following equations, for 2 g l$^{-1}$ fumarate concentration:

$$r_S(S) = \frac{1}{Y_{XS}} \frac{\mu_{\max} S}{K_S + S}.$$

$$r_O(S) = \frac{1}{Y_{OS}} \frac{\mu_{\max} S}{K_S + S}.$$

$$r_c(S) = \frac{1}{Y_{CS}} \frac{\mu_{\max} S}{K_S + S}.$$

### Cell volume determination

Cell volume was determined by microscopy imaging of live *E. coli* expressing a photo-switchable EYFP. LacY-EYFP fusion expression from pBAD-LacY-EYFP plasmid was induced in various ways, depending on the condition, in which cells were grown. In case of glucose-growing cells, this was done with addition of 0.02% arabinose 12 h prior to the measurements. This induction was enough to ensure high enough levels of LacY-EYFP expression in cells that were subsequently switched to medium with fumarate or medium without a carbon source, without further induction in the new conditions. In fumarate-growing cells, no induction was necessary and the leaky expression of protein was sufficient for imaging. Microscope imaging was performed as described before (Biteen *et al*, 2008). Localization of the LacY protein indicating the localization of cell membrane was digitized by manually selecting pixels on one side of longer cell axis, using ImageJ. The point coordinates were then transformed in the XY coordinate system, so the longer cell axis laid on the *x*-axis of the system. Simpson's rule was used to calculate individual cell volumes equal to the volume of a solid of revolution (i.e. a solid created by rotating the transformed points around the *x*-axis).

### ppGpp concentration determination

ppGpp was quantified as described before (Traxler *et al*, 2008), with slight modifications. Cultures were prepared as for fumarate uptake rate determination. At least $2 \times 10^{10}$ cells (with the exact number determined with flow cytometry) were harvested by removing the culture medium through fast filtration (< 60 s, 0.2-µm nylon membrane filter, Sigma) and placing the filter with cells immediately (< 5 s) into 3 ml of 1 M formic acid pre-cooled to 0°C. The filters containing *E. coli* cells were rinsed by pipetting the formic acid over the side of the filter containing the cells until all the cells were resuspended, and cells were further incubated without the filter for 30–60 min in the formic acid at 0°C. After that, the cell suspension

was centrifuged (1 min, 21,000 *g*, 4°C) and the cell pellet was discarded. Water and formic acid were removed by freeze-drying overnight. The dried metabolites were then resuspended in 0.13 ml of 0.1 M formic acid by vortexing and ultrasonication, centrifuged (1 min, 21,000 *g*, 4°C), filtered through a Spin-X centrifuge filter (Corning, 0.22-µm nylon membrane, 3 s, 4,000 *g*), and analyzed using a HPLC-UV method with absorbance measurement at 260 nm. A PL-SAX anion exchange column (Agilent) was used at a column temperature of 60°C. A 35-min linear gradient of two solvents was employed: 0 min: 100% A; 30 min: 0% A; 33 min: 0% A; 33 min: 100% A, where A: 0.01 M K$_2$HPO$_4$, pH 2.6, B: 0.5 M K$_2$HPO$_4$, pH 3.5. The flow rate was 1 ml min$^{-1}$. Peaks were quantified based on a ppGpp standard curve (Jena Biosciences) prepared in 0.1 M formic acid, and the intracellular concentrations were calculated considering the number of cells harvested and the determined cell volumes.

### Metabolite concentration determination with LC-MS/MS

Cultures were prepared as for fumarate uptake rate determination. At least $2 \times 10^9$ cells (with the exact number determined with flow cytometry) were harvested by removing the medium through fast filtration (< 60 s, 0.2-µm nylon membrane filter, Sigma) and placing the filters with cells immediately (< 5 s) into 3 ml of acetonitrile, methanol, water, and formic acid (40:40:19.9:0.1 by volume) pre-cooled to −20°C and with U$^{13}$C internal standard (see below) spiked in. The filters containing *E. coli* cells were then incubated for 60–120 min in the mixture at −20°C and, at the end of incubation time, rinsed by pipetting the extraction mixture over the side of the filter containing cells, until all the cells were removed from the filter. The mixture was gathered and centrifuged (10 min, 21,000 *g*, 4°C). Organic solvents were removed from the supernatant in a vacuum concentrator (2 h, RT), and the remaining water was removed by freeze-drying overnight. The samples were then resuspended in water by vortexing and ultrasonication, centrifuged (2 min, 21,000 *g*, 4°C), and analyzed using a UHPLC-MS/MS method (injection volume of 10 µl). Quantification was done by relating the samples' $^{12}$C/$^{13}$C peak area ratios to calibration curves constructed using pure $^{12}$C standard in conjunction with the global U$^{13}$C internal standard. The UHPLC-MS/MS system consisted of a Dionex Ultimate 3000 RS UHPLC (Dionex, Germering, Germany) with the sample compartment permanently cooled to 4°C, a Waters Acquity UPLC HSS T3 column with precolumn (dimensions: 150 × 2.1 mm, particle size: 3 µm; Waters, Milford, MA, USA) was used at 50°C column temperature. A linear binary UHPLC gradient was employed: 0 min: 100% A; 5 min: 100% A 10 min: 98% A; 11 min: 91% A; 16 min: 91% A; 18 min: 75% A, 22 min: 75% A; 22 min: 0% A; 26 min: 0% A; 26 min: 100% A; 30 min: 100% A, where solvent A was composed of water:MeOH 95:5, 10 mM tributylamine (2.4 ml per l), 15 mM acetic acid (0.86 ml per l), and 1 mM 3,5-heptanedione (0.237 ml per l), and solvent B was isopropanol. The flow rate was 0.35 ml min$^{-1}$. Quantification was done via multiple reaction monitoring on a MDS Sciex API365 tandem mass spectrometer upgraded to EP10+ (Ionics, Bolton, Ontario, Canada) and equipped with a Turbo-Ionspray source (MDS Sciex, Nieuwerkerk aan den Ijssel, Netherlands). The source parameters were as following: NEB (nebulizing gas, N$_2$): 12 a.u., CUR (curtain gas, N$_2$): 12 a.u., CAD (collision-activated dissociation gas): 4 a.u., IS (ion spray voltage):

−4,500 V, TEM (temperature): 500°C. If concentrations were measured in multiple campaigns, the data were analyzed with a mixed effects model using the campaign as a nuisance variable. Otherwise, mean and standard error were calculated normally. The $^{13}$C internal standard was prepared by growing *Saccharomyces cerevisiae* on $^{13}$C uniformly labeled glucose (Cambridge Isotope Labs) as a sole carbon source followed by quenching and metabolite extraction as described before (Siegel *et al*, 2014).

**Absolute protein quantification**

About $1.5 \times 10^9$ growing, persister, or starved cells were centrifuged (1 min, 16,000 *g*, 4°C), washed twice with ice-cold PBS, and the cell pellet was frozen in liquid nitrogen. Cell pellets were lysed in 50 µl lysis buffer (2% sodium doxycholate, 0.1 M ammonium bicarbonate) and disrupted by two cycles of sonication for 20 s (Hielscher ultrasonicator). Protein concentration was determined by BCA assay (Thermo Fisher Scientific) using a small sample aliquot. Proteins were reduced with 5 mM TCEP for 15 min at 95°C, alkylated with 10 mM iodoacetamide for 30 min in the dark at room temperature, and quenched with 12.5 mM *N*-acetylcysteine. Samples were diluted with 0.1 M ammonium bicarbonate solution to a final concentration of 1% sodium doxycholate before digestion with trypsin (Promega) at 37°C overnight (protein to trypsin ratio: 50:1). After digestion, the samples were supplemented with TFA to a final concentration of 0.5% and HCl to a final concentration of 50 mM. Precipitated sodium doxycholate was removed by centrifugation (15 min at 4°C at 21,000 *g*). Then, peptides were desalted on C18 reversed phase spin columns according to the manufacturer's instructions (Macrospin, Harvard Apparatus), dried under vacuum, and stored at −80°C until further processing.

About 1 µg of peptides of each sample was subjected to LC–MS analysis using a dual pressure LTQ-Orbitrap Velos mass spectrometer connected to an electrospray ion source (both Thermo Fisher Scientific) as described recently (Glatter *et al*, 2012) with a few modifications. In brief, peptide separation was carried out using an EASY nLC-1000 system (Thermo Fisher Scientific) equipped with a RP-HPLC column (75 µm × 45 cm) packed in-house with C18 resin (ReproSil-Pur C18–AQ, 1.9 µm resin; Dr. Maisch GmbH, Ammerbuch-Entringen, Germany) using a linear gradient from 95% solvent A (0.15% formic acid, 2% acetonitrile) and 5% solvent B (98% acetonitrile, 0.15% formic acid) to 28% solvent B over 90 min at a flow rate of 0.2 µl min$^{-1}$. The data acquisition mode was set to obtain one high-resolution MS scan in the FT part of the mass spectrometer at a resolution of 120,000 full width at half maximum (at *m/z* 400) followed by MS/MS scans in the linear ion trap of the 20 most intense ions. The charged state screening modus was enabled to exclude unassigned and singly charged ions and the dynamic exclusion duration was set to 20 s. The ion accumulation time was set to 300 ms (MS) and 50 ms (MS/MS).

For label-free quantification, the generated raw files were imported into the Progenesis LC-MS software (Nonlinear Dynamics, Version 4.0) and analyzed using the default parameter settings. MS/MS-data were exported directly from Progenesis LC-MS in mgf format and searched against a decoy database of the forward and reverse sequences of the predicted proteome from *E. coli* (Uniprot, download date: 15/6/2012, total of 10,388 entries) using MASCOT. The search criteria were set as following:

Full tryptic specificity was required (cleavage after lysine or arginine residues); three missed cleavages were allowed; carbamidomethylation (C) was set as fixed modification; oxidation (M) as variable modification. The mass tolerance was set to 10 ppm for precursor ions and 0.6 Da for fragment ions. Results from the database search were imported into Progenesis, and the protein false discovery rate (FDR) was set to 1% using the number of reverse hits in the dataset. The final protein lists containing the summed peak areas of all identified peptides for each protein, respectively, were exported from Progenesis LC-MS and further statically analyzed using an in-house developed R script (SafeQuant) (Glatter *et al*, 2012).

**Principal component analysis and GOterm enrichment analysis**

First, we applied two-dimensional principal component analysis (without scaling) using the FactoMineR package for R to datasets describing relative fold difference in protein concentrations between all the analyzed conditions (14 conditions or three conditions at the same time). PCA assigned two coordinates (one for each dimension) based on correlation coefficients calculated between each and every of the analyzed proteomes. These coordinates placed each of the analyzed proteomes in a two-dimensional space. Proteomes not generated in this study were then projected on the two-dimensional PCA space. This PCA space was generated based on the proteomes measured in this study but only using a subset of proteins that measured both in this and the external study. Furthermore, for each protein and each of the two new PCA dimensions, a correlation coefficient was calculated between (1) a numerical vector containing the particular protein concentration fold-change values between each of the analyzed proteomes and (2) a numerical vector containing the coordinate along the particular PCA dimension of each of the analyzed proteomes. These correlation coefficients indicate whether an increase or decrease in each protein concentration correlates with the position of the proteome along the dimensions of the PCA space. Along each correlation coefficient, a *P*-value indicating whether this correlation coefficient is significantly different from 0 was calculated. Then, we created (for each of the two PCA dimensions) two lists containing the protein names and *P*-values of their positive or negative correlation coefficients, thus enabling us to distinguish to which direction of the PCA dimension they contribute (the negatively correlated proteins contribute in an opposite direction than the positively correlated proteins). Then, these lists were supplemented with protein names that do not contribute to the particular direction of the PCA dimension (i.e. a list with proteins positively correlated with dimension 1 was supplemented with a list of proteins negatively correlated with dimension 1), with *P*-value of these added proteins set to 1. This supplementation was done in order to include all the measured proteins in GOterm enrichment analysis described below.

GOterm enrichment analysis was performed using the TopGO package for R. GOterm-gene annotations were downloaded from Bioconductor (the org.Eck12.e.g.db database). A protein was selected as significant if the *P*-value of its correlation coefficient (see above) was lower than 0.1. Then, GOterms were assigned using the elim algorithm, which walks through GOterm hierarchy from the highest to the lowest level of detail, eliminating broad GOterms given that more detailed, child GOterms were selected. For every

GOterm, we determined significance of its enrichment using the GOtTest function (which considered the number of significant proteins including their *P*-values), and thereby, we created a ranked list of the enriched GOterms.

### Enrichment analyses of sigma factor, transcription factor, and sRNA activity

Activity of sigma factors, transcription factors, and sRNA was determined as previously described with small adaptations (Zampar *et al*, 2013). Specifically, the subsets of regulated proteins were selected by clustering performed with STEM software (version 1.3.8) (Ernst & Bar-Joseph, 2006), which first generated a set of random model expression profiles (representing protein concentration change over time). Then, protein expression profiles were assigned to the random model expression profiles based on their correlation to these profiles. Random model expression profiles with statistically significant number of protein profiles assigned to them were then clustered depending on how well they correlated with each other. This resulted in lists of proteins that changed their concentrations significantly during the time course, clustered along similar expression profiles. The settings used for analysis can be found in Appendix Table S8. For enrichment analysis of sigma factor activity, we used sigma factor–gene associations, transcription factor–gene associations, or sRNA-gene associations from the RegulonDB database which were supported by experimental evidence. For each tested factor *j*, a *P*-value based on a hypergeometric mean was calculated:

$$p_j = 1 - \sum_{i=0}^{k_j-1} \frac{\binom{M_j}{i}\binom{N-M_j}{n-i}}{\binom{N}{n}},$$

where *N* is the total number of proteins, *n*—number of proteins in the investigated subset, $M_j$—number of proteins associated with sigma factor *j* in the complete protein list, $k_j$—number of proteins associated with sigma factor *j* in the investigated subset. The *P*-value, after False Discovery Rate adjustment, is equal to the probability that the observed protein expression change occurred by chance and not because of a real regulatory effect of the investigated sigma factor.

### Determination of fraction of persisters and growing cells

Cell were switched from M9 medium with glucose to M9 medium with fumarate as described previously, with staining (Kotte *et al*, 2014). Staining was performed using the PKH-67 dye (Sigma-Aldrich). Multiple cell count and fluorescence intensity measurements were taken during the culturing period. The fraction of growing and non-growing cells after the shift was determined by fitting a model of two Gaussian distributions and of exponential growth to the measured fluorescence intensity, as described before (Kotte *et al*, 2014).

### Estimation of maximum ATP production rate with flux balance analysis

A stoichiometric genome-scale metabolic network model of *E. coli* metabolism (Reed *et al*, 2003) was constrained with the measured extracellular rates (carbon uptake, gas exchange, and biomass production) within their 99.5% confidence intervals. The units of measured rates were converted from [fmol cell$^{-1}$ h$^{-1}$] to [mmol gDW$^{-1}$ h$^{-1}$] using the measured cell volumes and assuming that 0.74 of cell mass is water (Ishkawa *et al*, 1995) and that the average cell density is 1.105 kg l$^{-1}$. The model was solved using GAMS with maximization of ATP hydrolysis rate as the objective.

### Accession codes

All mass spectrometry raw data files have been deposited to the ProteomeXchange Consortium (http://proteomecentral.proteomexchange.org) (identifier PXD001968).

**Expanded View** for this article is available online.

### Acknowledgements

We thank E. Wit and B. Niebel for help with statistical analysis; A. van Dam and H. Permentier for help with metabolite quantification; A. Robinson for help with microscopy; K. Gerdes, E. Maisonneuve, M. Askvad Sørensen, A. Papagiannakis, K.E.S. Leupold, and T. Kimkes for discussions and critically reviewing an earlier version of the manuscript, and H. Schramke for discussions and critically reviewing this manuscript. We also thank members of the MSB group for support. This work was partly financed by the Netherlands Organisation for Scientific Research (NWO) through a VIDI grant to MH (project number 864.11.001).

### Author contributions

JLR and MH conceived and designed the study, and wrote the manuscript with input from all authors. JLR performed experiments (antibiotic tolerance with SV, proteomics with AS, metabolite measurements with DS) and analyzed the data. ÁDO performed transcript quantification.

### Conflict of interest

The authors declare that they have no conflict of interest.

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
