## [Review Process File · Molecular Systems Biology]

Bacterial persistence is an active σ^S stress response to metabolic flux limitation

Jakub Leszek Radzikowski, Silke Vedelaar, David Siegel, Álvaro Dario Ortega, Alexander Schmidt and Matthias Heinemann

Corresponding author: Matthias Heinemann, University of Groningen

Review timeline:

Submission date:	08 April 2016
Editorial Decision:	31 May 2016
Third delayed report:	02 June 2016
Revision received:	12 August 2016
Editorial Decision:	18 August 2016
Revision received:	19 August 2016
Accepted:	24 August 2016

Editor: Thomas Lemberger

Transaction Report:

1st Editorial Decision

31 May 2016

We have now heard back from two out of the three referees who agreed to evaluate your manuscript. Given that the overall recommendations provided by the two reviewers are similar, I prefer to make a decision now rather than delaying further the process. As you will see from the reports below, the referees find the topic of your study of potential interest. They raise, however, several points that should be convincingly addressed in revision:

- One of the major issues raised is the need to provide a better characterization of the (phenotypic) similarity between the 'glucose-to-fumarate' persisters described in this study and 'classical' persisters obtained in rich conditions.
- The results of the proteomics PCA analysis should also be clarified and the contribution of change in growth rate to the changes observed at the proteome level should be analyzed.

REFeree COMMENTS

Reviewer #1:

In this manuscript, Radzikowski and co-workers investigate the role of metabolism in persister bacteria. Persisters are generated through a nutrient-shift which results in a large fraction of non- or slow-growing cells which are distinct from growing cells and starved cells but resemble persisters (an antibiotic-tolerant subpopulation that occurs in clonal bacterial populations, usually due to a stochastic phenotypic switch that shifts a small fraction of the population to a non- or slow-growing

state). The authors use a wide range of methods and assays to establish that *Escherichia coli* bacteria shifted from glucose to fumarate share key hallmarks of persisters; they ultimately identify these cells as persisters. Their antibiotic tolerance phenotype provided the first key indication; further characterizations included toxin/antitoxin system expression, cell size, growth rate, energy state, proteomics (including comparison to the stringent response regulon), metabolite analysis and finally comparisons between several relevant mutant strains. A 'system-level' model based on a positive feedback loop generated via reduced growth rate which subsequently reduces transcription and translation is presented; in this model, positive feedback leads to bistability in metabolism which prevents escape from the non- or slow-growing state which is crucial for the occurrence of persistence. The authors argue that regulation through toxin-antitoxin-systems and the stringent response alone are not sufficient to achieve the observed large fraction of persisters. In contrast to starved cells, the persister cells (generated by the glucose-fumarate shift) continue to generate chemical energy at a high rate. This energy stems from fumarate which the persisters mostly do not use for biomass generation (as fumarate-adapted cells would) but rather for ATP production. This energy can be invested into survival and maintenance (potentially to achieve outgrowth at a later time point) and might also be invested in antibiotic resistance mechanisms (such as ATP-dependent drug efflux mechanisms).

Bacterial persistence is an important phenomenon both from a basic research perspective and from an applied point of view. Thus, an improved characterization of the phenotypic state of persister bacteria, their metabolism and proteome as performed in this work would certainly be of great interest for a broad audience of systems biologists and microbiologists. The present work provides relevant new data and would present a considerable technical advance, provided that the protocol used to generate the persisters does not introduce any artifacts. This 'new' way of making persisters would indeed enable the use of experimental techniques where large number of cells are needed to study persisters. The work seems technically sound and the main text is overall comprehensible, but the presentation could certainly be improved and shortened (see specific issues below). Further, it is not clear if the model presented in Fig. 7 goes substantially beyond the state-of-the-art; its role in the context of previous work needs to be clarified (see below). Overall, this work is of interest and could be considered for publication, provided that the authors can convincingly address the following issues.

Major issues:

1. A central issue is how similar the cells called 'persisters' by the authors really are to spontaneous persisters that most people will think of when they hear this term. The authors investigate starved cells in parallel in an attempt to show the similarities and differences to persisters. Although this analysis is extensive and there are similarities and differences visible in the data, it is often (especially in the introduction and first couple of results sections) not clear why these phenotypes are compared. There are other methods to enrich the persister cells (occurring in nutrient rich conditions) which could then be investigated with some of the methods used here, to further strengthen the claim that the fumarate-shifted cells indeed resemble persisters rather than just starved cells. Such a validation would be extremely helpful, at least for some of the simpler phenotypes. A closely related question is if such a persistent state would also occur for shifts to other carbon sources. Or would virtually any transient lowering of growth rate yield the same fraction of persister cells (e.g. are the cells that transiently stop growing in a classical diauxic shift from glucose to lactose also persisters)?
2. The model of persistence does not appear entirely novel and needs to be properly placed into the context of recent work. The conceptual model presented in Fig. 7 consists mainly of a positive feedback loop that leads to bistability, pushing cells into growth arrest after a perturbation. Transcription and translation decrease and drive the cells into a non-growing state hence running the 'vicious cycle'. In that sense the proposed model seems conceptually almost identical to that described by Klumpp et al. (*Cell*, 2009) who show that bistability (co-existence of growing and non-growing cells) occurs via such global effects (growth-dependent feedback). In that paper, the expression level of proteins (i.e. lower levels e.g. due to lower transcription/translation) results in a lowering of the growth rate and such a feedback occurs, leading to the occurrence of persisters in a population. Similar ideas have also been presented in other recent papers. It will be important to put the present work carefully into the context of this prior work and clearly highlight how it goes beyond the state-of-the-art.

3. How were the antibiotics used for the test in Fig. 1 chosen? E.g. it is well-established that slow- or non-growing bacteria are generally not killed by ampicillin (or other beta-lactams). Chloramphenicol should be bacteriostatic (i.e. it should not kill bacteria but merely inhibit growth, even at high concentrations); hence, it is not clear what we learn from it in Figure 1B. It would help to explain clearly why experiments using these antibiotics are needed and what we learn from them with respect to the persister phenotype.

4. In the proteome analysis, the role of growth rate changes alone needs to be clarified. Cells between 2 and 8 hours after the fumarate-shift are hardly growing (growth rate of 0.02 per hour), but fumarate-adapted (and glucose-adapted) cells are growing faster. It is well-established that the global proteome is strongly affected by growth rate; thus, it is entirely expected that these proteomic states are different (and those of the growing cells more similar, see Fig. 4). The observed similarity of the proteomic states between starvation-shifted cells and the fumarate-shifted cells within the first 2 hours might primarily stem from the fact that during this time window they have similar growth rates (Fig. 2B). Also the directionality shown in Fig. 4B and mentioned in lines 250-256 might be a general effect coming from growth rate reduction. The authors should clarify this point. If they want to make a claim that the glucose-fumarate shift leads to a proteomic state that is in some sense unusual, it would be important to perform a similar analysis for a different nutrient shift as reference.

5. The finding of higher levels of proteins involved in stopping transcription and translation (RNases, RMF) (lines 308-318) in persisters indeed suggests an active response that initiates and supports the persister state. The comparison of persisters to starved cells, however, yields much weaker differences; the only clearly visible difference is for RMF (and that only at the later time points, as the other differences seem to be insignificant). However, at later time points, the fumarate-shifted cells approach adaptation to fumarate (i.e. it starts getting utilized for biomass growth), and the starved cells inflate and their cell count slightly decreases (Fig. 1). Hence, major reorganization is already taking place at this time point in both cell types. It should be made clearer what can be learned from the comparison of persisters vs. the starved cells with respect to the selected (Fig. 5B, C) proteins. Some estimates of error and significance in Fig. 5B,C would also be helpful (some of the differences shown fluctuate strongly among the time points and seem insignificant, e.g. RNase I in both panels).

Other issues:

1. The structure of the introduction could be improved. In particular, the non-/slow-growing cells under investigation here are termed persisters in the introduction relatively early, although a previous publication from the same group (Kotte et al. 2014) specifically avoided using that term. However, a goal of the present study seems to be to prove that these cells are in fact persisters. That goal seems to be at least partly achieved during the first part of the results section; however the structure of the introduction does not reflect that order.

2. With respect to paragraph line 94 to 103 and also line 89: A clear motivation to compare the antibiotic susceptibility of starvation-shifted cells with the fumarate-shifted cells is missing. Further, it would help to clarify here if the authors suggest that there is a general (and identical) mechanism present in both types of cells that allows resistance to most antibiotics, or not. Regarding the assay of determining the fractions of surviving cells: are fractions larger than 1 the result of the normalization to a noisy measurement and non-recovering cell level?

3. Why are fumarate and O₂ uptake rate and CO₂ production rate for the starved cells not reported (Fig. 2 and Table S2)?

4. On the one hand it is revealed that the yield of ATP per fumarate is high in the persisters and it is shown that fumarate is invested into energy production rather than biomass production. On the other hand it is calculated that the achieved absolute ATP-production rate is exactly sufficient for the so called non-growth-maintenance. Nonetheless, the cells do grow at a rate of 0.02 per hour (Fig. 1). Please clarify that this is feasible (potentially using the range presented for the non-growth-maintenance).

5. Line 213-216: Drug-tolerance is also shown for the starvation-shifted cells, and long-term survival is not shown (or mentioned) for either. Hence, the conclusion made here does not clarify the (dis)similarities found here between the two types of cells that were investigated.

6. It is not presented clearly how the authors come to the conclusions made in line 273-277; the enrichment analysis (Fig. 5 and Suppl. Table 4) indicates that only 11 out of 20 of the GO-terms found in both conditions actually overlap.

7. The conclusions made in lines 347-354 need a bit more explanation. What is the fraction of the whole sigma S regulon that is up-regulated during starvation? What is the overlap with the persisters?

8. The central carbon metabolite analysis (Fig. 6) implies that the levels of ATP are the same as for glucose-growing, fumarate-growing and fumarate-shifted cells (all normalized to the glucose-growing cells). However, earlier in the results section (Fig. 3) it is shown that ATP yield (ATP/fumarate) and ATP production rate (mmol/gDW/h) is higher in the persisters than in the fumarate-growing cells. Together with the time series of the metabolite data from the starved cells this implies that ATP levels are similar over the different conditions, but ADP and AMP are not (Fig. 5). AMP levels for starved cells are highest directly after the shift, and remain moderately high for the rest of the observed time. It would help to clarify how those patterns relate to the finding presented in Fig. 3E, which shows a gradually lowering in energy charge for the starved cells.

9. Considering the schematic model presented in Fig. 7, it should be noted that ppGpp influences translation not only via the stringent response and the T/A-systems, but also directly through RMF (storing of deactivated ribosomes). ppGpp can thus be considered a regulator of the 'system-level' aspect presented here because an increase in ppGpp generally reduces growth rate (making fewer ribosomes available for translation).

10. It is unclear if the landscape in Fig. 8 is the result of a quantitative model or a schematic. Does the phenotypic instability (Z-axis) quantify the fractions persisters vs. growing cells in the whole population?

Reviewer #3:

The work of Radzikowski et al. compares the metabolism of non-growing cells mainly in two different conditions:

1. Shift from glucose to fumarate that results in growth arrest of most of the population and growth of a minority
2. Shift to no carbon source medium: cells are abruptly starved and do not grow

As expected, both conditions results in slower growth or growth arrest and therefore high tolerance of the whole population to several antibiotic drugs. Not surprisingly, both non-growing states are similar also in their expression levels of several TA systems and differ from glucose growing cells. Interestingly, they do find two drugs (CCCP and ofloxacin) that result in higher tolerance of the fumarate cells than the starved cells. The authors conclude that tolerance in the fumarate culture differs from tolerance due to starvation.

Focusing on the tolerance induced by the shift to fumarate, the authors show that these persisters are metabolically active and have a distinct metabolism when compared to cells growing on fumarate. By measuring the rate of fumarate uptake compared to the rate of oxygen uptake, they conclude that the ATP production is more efficient in persisters.

The core of the results in this work comes mainly from the extensive proteomic analyses of cells in various conditions:

Persisters cells as defined above (shift glucose to fumarate), starved cells (shift glucose to null, fumarate to null), exponential fumarate, exponential glucose. This extensive data set enables the authors to see a general trend upon starvation that is followed upon all transitions within the PCA space, probably marking the main effect of growth reduction.

The authors then turn on specific proteins to understand what characterizes the proteome of persistent cells compared to starved cells and conclude that sigmaS is the major factor shaping the persister proteome. They categorize the proteins that are characteristic of the different states. Finally, they show that rpoS is essential for triggering the growth arrest, whereas deletion of 10 TA modules has little effect.

This extensive analysis is then summarized in a new view of how persistence may be understood. The results presented here should be extremely interesting to microbiologists interested in the physiology of bacteria outside the exponential growth. The analysis includes here the transitions between carefully chosen states and their time-dependence and should shed light on the physiology on non-growing bacteria and their resilience to antibiotic treatment.

Minor comments:

- In Fig. 2A Cell size evaluation was done by super resolution microscopy and image analysis: it would be good to add images showing the typical differences in cell size at different time-points.
- In Fig. 2C-E the legend states the persisters cells are measured but it is not clear what is plotted. What are the different grey shaded points? Are they still representing the two different conditions as in B?
- How can the authors evaluate the contribution of the 1% of fumarate growing cells to metabolism of the whole culture? How can the metabolism be attributed to persisters?
- The measured rate of oxygen and fumarate uptake do not show a steady state behavior. The graphs start increasing only towards the end. How can rates be extracted and steady-state analysis be done?
- in p. 10 the authors write that "the sum of adenylate nucleotides concentration is constant in persisters (Fig. 3F) but there is a 50% decrease in Fig. 3F.
- Fig 4 legend: it is written that the distance between points is inversely correlated with similarity between proteomes.
- Fig. 4 legend: Just to make the legend clearer: "circle" is usually for an open symbol (as in geometry) whereas "disk" is used for the filled symbol.
- The authors open the results with selected measurements of TA modules in persisters. How do these results concur with the proteome analyses? According to the little effect of the 10TAs deletion on their assay, it seems that TAs are not important in the type of persisters studied here.
- in p.21-22 the authors mention several times the "persistence level" of strains with and without rpoS changes but the figure shows only data on growth/no growth and not survival to antibiotics. As persistence is here a central theme, they should show survival assays of the mutants in Fig. 7B to one of the drugs tested in Fig. 1B.
- The view of persistence and growth as two attractors of the metabolic fluxes is appealing. However, it is not clear what distinguishes "persistence" in this landscape from simply a growth/no growth landscape that is driven by starvation.

Third delayed report

02 June 2016

We have now finally received the third delayed review of your manuscript (see below). This reviewer is globally supportive and makes suggestions that we would kindly ask you to carefully address in the revision.

 REFEREE COMMENTS

Reviewer #2

The work presented here is a continuation of the elegant work by Kotte et al. 2014 where the Authors used the same experimental set-up to analyse phenotypic bistability during sudden carbon source shifts. Here they address the question of bacterial persistence induced by sudden carbon source shifts. The manuscript presents a huge amount of very carefully obtained data. However the logic, in particular, in the last part of Results, is difficult to follow. Most importantly, the Authors invoke ppGpp and SpoT in their model without presenting data to support their claim. They show strong data proposing that the sigma starvation factor RpoS is central to generating persisters. Fig 7A (the model) must be modified to reflect this fact or data supporting the involvement of SpoT should be added. In the same vein, the Authors seemingly use stringent response (mediated by

ppGpp) and the general stress response (mediated by RpoS) as being one and the same but this is not correct (see e.g. reviews by Susan Gottesman and or Regine Hengge) since even though the two stress response regulons overlap they certainly are yet very different. Another general criticism is that the Results section contains a lot of interpretations. This is confusing and should be removed. Some of these passages are mentioned below.

Specific points:

The title is too general and in my mind could invoke the finding that persister formation depends strongly on RpoS (Fig 7B, C). Alternatively "Bacterial persistence" could be replaced by "Responsive diversification" in the title.

Introduction:

28-29: reformulate sentence

47: Bacteria in macrophage vacuoles are starving and induce the stringent response to survive the starvation.

50-51: sentence is imprecise.

55-57 (important point): It is well-described in the literature that persisters are heterogeneous, see e.g. Amato & Brynildsen, Current Biol 2015. However, the papers by Maisonneuve et al., 2013 and Nguyen et al., 2011 show that the vast majority of persisters (up to 99% of persisters in growing populations) depend on ppGpp in both organisms and also on TAS in E. coli. Where the minor fraction (less than 1%) of persisters come from is difficult to analyse because the numbers are so low. Such persisters are generated by another yet-to-be-discovered mechanism and are probably not highly relevant in the present context. The text should be rephrased to reflect this fact.

64: the authors mention Amato & Brynildsen 2014 and Kotte et al., 2014 in the same sentence. However, the experimental models used by the two groups are very different. In the diauxic growth model used by Amato & Brynildsen 2014, the cells gradually shift from one carbon source to another, a situation that often occurs in Nature. The experimental approach used here and also by Kotte et al., 2014 invokes a sudden shift from one carbon source to another and in my mind it is questionable whether bacteria encounter such conditions in more realistic settings. Nevertheless, using this "abrupt shift model" Kotte et al., 2014 elegantly showed that abrupt shifts from glucose to gluconeogenic C-sources induce "responsive diversification" that depended on the transcription factor Cra, an obvious and important result. Moreover the proteome resemblance and energy differences of starved and down-shifted cells certainly argue that the model is physiologically relevant and useful. However, it might be appropriate to state that the multidrug tolerant persisters generated by sudden metabolic shifts are useful experimental approximations to more realistic settings as also argued by researchers using stationary phase cells for the same purpose.

Results:

Fig. S1A is very difficult if not impossible to understand. I had to go back to Fig 2C of Kotte et al., 2014 to understand what's going on. This makes the reading quite heavy.

97: "tolerance" should be "tolerant".

112-116: The sentence is incomprehensible and should be clarified and moved to the Discussion since it invokes an interpretation.

141-142: "very similar" is unclear. Please explain.

294; "on" should be "or".

292-307 is a mix of results and interpretations and is slightly confusion. Delete or move to Discussion.

349-351: Here is an inference that pops up out of the blue. The authors write: "As RpoS was up-regulated for the whole period of our observation (Sup. Tab 5), this suggested that the stringent response is sustained over time,". Here the authors again make the inaccuracy of equalizing the

RpoS dependent general stress response and the stringent response. There are no data in the manuscript supporting that "the stringent response is sustained over time" and it should either be substantiated or removed. See also my comment to the model in Fig 7A below.

374: here the authors suddenly discuss similarities of starvation patterns of pro- and eukaryotes.

375: same as above, no data to support the claim about the stringent response.

Fig 7A, the model. The article does not contain any data on ppGpp or SpoT and the model is therefore not supported by the data. SpoT and ppGpp should be removed or data to support the claim should be added.

440: The authors are too rigorous in the interpretation of their own data. Fig 7B certainly shows that RpoS is highly important for the generation of persisters. It is a well-known fact that persisters are generated by multiple parallel mechanisms so it is not to be expected that a single gene or pathway will account for all persisters in a given situation.

I suggest the authors focus the paper more on e.g. RpoS whose expression, of course, is under the control of ppGpp and thus the stringent response. But to postulate that persister formation in their model is controlled by SpoT and ppGpp (which might certainly be true) needs experimental support.

Based on these considerations I suggest that the manuscript should be returned for a "major revision" allowing for either substantial rewriting and modification of the model (Fig 7A) or addition of crucial experiments showing the involvement of SpoT and ppGpp in the formation of persisters in the model used by the authors.

1st Revision - authors' response

12 August 2016

Response letter

Editor:

- *One of the major issues raised is the need to provide a better characterization of the (phenotypic) similarity between the 'glucose-to-fumarate' persisters described in this study and 'classical' persisters obtained in rich conditions.*

- *The results of the proteomics PCA analysis should also be clarified and the contribution of change in growth rate to the changes observed at the proteome level should be analyzed.*

First, thank you very much for the excellent and very helpful reviews!

Second, as outlined below, we carefully addressed these points and also all the others raised by the reviewers.

Reviewer #1:

In this manuscript, Radzikowski and co-workers investigate the role of metabolism in persister bacteria. Persisters are generated through a nutrient-shift which results in a large fraction of non- or slow-growing cells which are distinct from growing cells and starved cells but resemble persisters (an antibiotic-tolerant subpopulation that occurs in clonal bacterial populations, usually due to a stochastic phenotypic switch that shifts a small fraction of the population to a non- or slow-growing state). The authors use a wide range of methods and assays to establish that Escherichia coli bacteria shifted from glucose to fumarate share key hallmarks of persisters; they ultimately identify these cells as persisters. Their antibiotic tolerance phenotype provided the first key indication; further characterizations included toxin/antitoxin system expression, cell size, growth rate, energy state, proteomics (including comparison to the stringent response regulon), metabolite analysis and finally comparisons between several relevant mutant strains. A 'system-level' model based on a positive feedback loop generated via reduced growth rate which subsequently reduces

transcription and translation is presented; in this model, positive feedback leads to bistability in metabolism which prevents escape from the non- or slow-growing state which is crucial for the occurrence of persistence. The authors argue that regulation through toxin-antitoxin-systems and the stringent response alone are not sufficient to achieve the observed large fraction of persisters. In contrast to starved cells, the persister cells (generated by the glucose-fumarate shift) continue to generate chemical energy at a high rate. This energy stems from fumarate which the persisters mostly do not use for biomass generation (as fumarate-adapted cells would) but rather for ATP production. This energy can be invested into survival and maintenance (potentially to achieve outgrowth at a later time point) and might also be invested in antibiotic resistance mechanisms (such as ATP-dependent drug efflux mechanisms).

Bacterial persistence is an important phenomenon both from a basic research perspective and from an applied point of view. Thus, an improved characterization of the phenotypic state of persister bacteria, their metabolism and proteome as performed in this work would certainly be of great interest for a broad audience of systems biologists and microbiologists. The present work provides relevant new data and would present a considerable technical advance, provided that the protocol used to generate the persisters does not introduce any artifacts. This 'new' way of making persisters would indeed enable the use of experimental techniques where large number of cells are needed to study persisters. The work seems technically sound and the main text is overall comprehensible, but the presentation could certainly be improved and shortened (see specific issues below). Further, it is not clear if the model presented in Fig. 7 goes substantially beyond the state-of-the-art; its role in the context of previous work needs to be clarified (see below). Overall, this work is of interest and could be considered for publication, provided that the authors can convincingly address the following issues.

Answer: Thanks for the excellent to-the-point summary of our work and the nice words about it.

In this revised version, we further compare (i) the persisters we generated in the “new” way, (ii) addressed the presentation issues, and (iii) clarified in how far the presented model goes beyond the state of the art. Furthermore, we have addressed the issues below as indicated.

Major issues:

1. A central issue is how similar the cells called 'persisters' by the authors really are to spontaneous persisters that most people will think of when they hear this term. The authors investigate starved cells in parallel in an attempt to show the similarities and differences to persisters. Although this analysis is extensive and there are similarities and differences visible in the data, it is often (especially in the introduction and first couple of results sections) not clear why these phenotypes are compared. There are other methods to enrich the persister cells (occurring in nutrient rich conditions) which could then be investigated with some of the methods used here, to further strengthen the claim that the fumarate-shifted cells indeed resemble persisters rather than just starved cells. Such a validation would be extremely helpful, at least for some of the simpler phenotypes. A closely related question is if such a persistent state would also occur for shifts to other carbon sources. Or would virtually any transient lowering of growth rate yield the same fraction of persister cells (e.g. are the cells that transiently stop growing in a classical diauxic shift from glucose to lactose also persisters)?

Answer: Here, the reviewer rose three points: (i) clarify in the introduction why persisters were compared with starved cells; (ii) compare “our” persisters with the “classical” ones; (iii) address whether similar persisters occur with other carbon source switches.

As for (i): we have completely rewritten the introduction. The previous introduction was indeed a bit convoluted. Now, we clearly introduce the different persister models and motivate our work in a much clearer way. Thanks for bringing up this point.

As for (ii): In addition to the similarities (between “our” persisters and the classical ones) that we already showed in the previous version (i.e. AB tolerance, upregulation of RpoS, upregulation of TAS), we now also show that ppGpp levels are increased in the persisters we generate.

As for (iii): In our earlier work (Kotte et al, 2014, MSB), we already showed that also during shifts to other carbon sources, a significant fraction of cells does not start to grow and thus does enter persistence. According to the model that we present in our current manuscript, any strong perturbation of metabolic homeostasis (i.e. significant drop in metabolic fluxes, for instance caused by nutrient change or by other stress) should lead to persistence. Along these lines, cells that are confronted with a glucose-to-lactose shift and which fail in restoring metabolic homeostasis would also enter the state of persistence. In fact, Amato and Brydnilsen, *Molecular Cell* 2013 (Fig. 1), demonstrated the existence of persisters when cells are subjected to diauxic shift from glucose to several different carbon sources.

2. The model of persistence does not appear entirely novel and needs to be properly placed into the context of recent work. The conceptual model presented in Fig. 7 consists mainly of a positive feedback loop that leads to bistability, pushing cells into growth arrest after a perturbation. Transcription and translation decrease and drive the cells into a non-growing state hence running the 'vicious cycle'. In that sense the proposed model seems conceptually almost identical to that described by Klumpp et al. (Cell, 2009) who show that bistability (co-existence of growing and non-growing cells) occurs via such global effects (growth-dependent feedback). In that paper, the expression level of proteins (i.e. lower levels e.g. due to lower transcription/translation) results in a lowering of the growth rate and such a feedback occurs, leading to the occurrence of persisters in a population. Similar ideas have also been presented in other recent papers. It will be important to put the present work carefully into the context of this prior work and clearly highlight how it goes beyond the state-of-the-art.

Answer: Please note that the model of Klumpp et al. evokes a protein, which directly affects growth (i.e. a toxic protein) and is then responsible to generate the growth bistability, because at low growth rates it is expressed at higher levels, leading to a further reduction in growth rate. Our model, in contrast, does not need such a toxic protein to generate the persisters. Thus, the two models are different. In this revised version, we improved the explanation of our model, and also explicitly highlight the difference between our model and the mechanisms described by Klumpp et al. to make sure that readers understand the difference.

3. How were the antibiotics used for the test in Fig. 1 chosen? E.g. it is well-established that slow- or non-growing bacteria are generally not killed by ampicillin (or other beta-lactams). Chloramphenicol should be bacteriostatic (i.e. it should not kill bacteria but merely inhibit growth, even at high concentrations); hence, it is not clear what we learn from it in Figure 1B. It would help to explain clearly why experiments using these antibiotics are needed and what we learn from them with respect to the persister phenotype.

Answer: The antibiotic concentrations, even for bacteriostatic antibiotics, were tested with fumarate-growing cells. There, even with bacteriostatic antibiotics, we have seen more than 4 hours of growth inhibition after removal of the antibiotic from the medium. Therefore, the growth resumption of persisters and starved cells is different from the one of fumarate-growing cells and thus a result of the phenotype of persisters/starved cells.

We agree that we did not properly motivate the chosen antibiotics. Ampicillin was chosen as the standard antibiotic to screen for stochastically-formed persisters as done in previous papers. As for the others, essentially, we chose antibiotics that target different cellular functions, hoping to elucidate differences in cellular activity between persisters and starved cells. We now mention this reasoning in the manuscript.

4. In the proteome analysis, the role of growth rate changes alone needs to be clarified. Cells between 2 and 8 hours after the fumarate-shift are hardly growing (growth rate of 0.02 per hour), but fumarate-adapted (and glucose-adapted) cells are growing faster. It is well-established that the global proteome is strongly affected by growth rate; thus, it is entirely expected that these proteomic states are different (and those of the growing cells more similar, see Fig. 4). The observed similarity of the proteomic states between starvation-shifted cells and the fumarate-shifted cells within the first 2 hours might primarily stem from the fact that during this time window they have similar growth rates (Fig. 2B). Also the directionality shown in Fig. 4B and mentioned in lines 250-256 might be a general effect coming from growth rate reduction. The authors should clarify this point. If they want to make a claim that the glucose-fumarate shift leads to a proteomic state that is in some sense

unusual, it would be important to perform a similar analysis for a different nutrient shift as reference.

Answer: Thank you very much for raising this point. Growth rate indeed can have a great influence on gene expression, and thus the shape of the proteome. To address this point, we now added an additional analysis using the proteome data from our recent large-scale screening (Schmidt et al, Nature Biotechnology, 2016). This analysis provided evidence that - next to the growth rate effects - there is still a distinct persister proteome, which is largely associated with stress response. We feel that this additional analysis significantly adds to the manuscript. We are very grateful for this reviewer having drawn our attention do this (admittedly) overlook point.

5. The finding of higher levels of proteins involved in stopping transcription and translation (RNases, RMF) (lines 308-318) in persisters indeed suggests an active response that initiates and supports the persister state. The comparison of persisters to starved cells, however, yields much weaker differences; the only clearly visible difference is for RMF (and that only at the later time points, as the other differences seem to be insignificant). However, at later time points, the fumarate-shifted cells approach adaptation to fumarate (i.e. it starts getting utilized for biomass growth), and the starved cells inflate and their cell count slightly decreases (Fig. 1). Hence, major reorganization is already taking place at this time point in both cell types. It should be made clearer what can be learned from the comparison of persisters vs. the starved cells with respect to the selected (Fig. 5B, C) proteins. Some estimates of error and significance in Fig. 5B,C would also be helpful (some of the differences shown fluctuate strongly among the time points and seem insignificant, e.g. RNase I in both panels).

Answer: The reviewer has a valid point asking whether the RMF differences that we report between the persisters and starved cells are significant, or whether it could be merely a result of other processes (such as inflation of starved cells). As for the inflation, note that because the protein expression data we report for each protein is relative to the rest of the proteome, an eventually diluted proteome due to cell inflation could not generate the observed differences. Still, to provide more confidence that the observed differences are significant, we now added error bars that we calculated based on the variation between the technical replicates. The variation between biological replicates, measured for glucose proteome, was similar to the variation between technical replicates in other conditions. Therefore, the variation seen in the results comes mostly from the method used.

Other issues:

1. The structure of the introduction could be improved. In particular, the non-/slow-growing cells under investigation here are termed persisters in the introduction relatively early, although a previous publication from the same group (Kotte et al. 2014) specifically avoided using that term. However, a goal of the present study seems to be to prove that these cells are in fact persisters. That goal seems to be at least partly achieved during the first part of the results section; however the structure of the introduction does not reflect that order.

Answer: As mentioned above, we have completely rewritten the introduction, which now has a much clearer structure and the goals of the study are much better presented. (Please note that in our Kotte 2014 paper, we already have called these cells “persisters”.)

2. With respect to paragraph line 94 to 103 and also line 89: A clear motivation to compare the antibiotic susceptibility of starvation-shifted cells with the fumarate-shifted cells is missing. Further, it would help to clarify here if the authors suggest that there is a general (and identical) mechanism present in both types of cells that allows resistance to most antibiotics, or not. Regarding the assay of determining the fractions of surviving cells: are fractions larger than 1 the result of the normalization to a noisy measurement and non-recovering cell level?

Answer: Starved cells have been used as a model for persistence in the past, and this is why we use them as a reference throughout the paper. While the general mechanism of antibiotic tolerance is most probably the slow-/no-growth of both starved cells and persisters, our analysis revealed some differences suggesting an active response. We have made both points clearer in our manuscript.

As for the technical question: Yes, this is the result of the normalization accounting for the inherent noise in the flow cytometry measurement (for instance, caused by particles smaller than 0.22 μ m present in the growth medium and electronic noise) and the number of cells that do not recover after the shift without any antibiotic treatment (i.e. dead cells). This aspect is mentioned in the method section.

3. *Why are fumarate and O₂ uptake rate and CO₂ production rate for the starved cells not reported (Fig. 2 and Table S2)?*

Answer: To generate the starved cells, we shifted glucose-grown cells to a medium without carbon source (meaning that there was no fumarate present, and thus fumarate uptake could not be determined).

As for the O₂ uptake or CO₂ production: We did measure these rates. However, the measured values were below the quantitative capabilities of the measurement system used and as such we did not report these values.

4. *On the one hand it is revealed that the yield of ATP per fumarate is high in the persisters and it is shown that fumarate is invested into energy production rather than biomass production. On the other hand it is calculated that the achieved absolute ATP-production rate is exactly sufficient for the so called non-growth-maintenance. Nonetheless, the cells do grow at a rate of 0.02 per hour (Fig. 1). Please clarify that this is feasible (potentially using the range presented for the non-growth-maintenance).*

Answer: Please note that we did not report the *total* (“absolute”) ATP production rate, but the ATP production rate that would be available (*as a surplus*) once the ATP required to sustain the 0.02 growth rate is subtracted. Specifically, the biomass equation (i.e. the reaction in the model that synthesizes biomass and equals to the growth rate) includes an ATP requirement equal to the amount of ATP needed to synthesize the biomass. Thus, the ATP needed to generate biomass at 0.02 rate was accounted for, and not included in the value we report. We have made this point clearer in the manuscript.

5. *Line 213-216: Drug-tolerance is also shown for the starvation-shifted cells, and long-term survival is not shown (or mentioned) for either. Hence, the conclusion made here does not clarify the (dis)similarities found here between the two types of cells that were investigated.*

Answer: Indeed, the conclusion about long-term survival was far-fetched and we removed it.

6. *It is not presented clearly how the authors come to the conclusions made in line 273-277; the enrichment analysis (Fig. 5 and Suppl. Table 4) indicates that only 11 out of 20 of the GO-terms found in both conditions actually overlap.*

Answer: In the previous version it was not clear that this statement was supposed to consider only the enhanced (i.e. up-regulated) functions. We have made this point clearer in the manuscript.

7. *The conclusions made in lines 347-354 need a bit more explanation. What is the fraction of the whole sigma S regulon that is up-regulated during starvation? What is the overlap with the persisters?*

Answer: We have added this information to the manuscript.

8. *The central carbon metabolite analysis (Fig. 6) implies that the levels of ATP are the same as for glucose-growing, fumarate-growing and fumarate-shifted cells (all normalized to the glucose-growing cells). However, earlier in the results section (Fig. 3) it is shown that ATP yield (ATP/fumarate) and ATP production rate (mmol/gDW/h) is higher in the persisters than in the fumarate-growing cells. Together with the time series of the metabolite data from the starved cells this implies that ATP levels are similar over the different conditions, but ADP and AMP are not (Fig. 5). [Comment added by the authors: We assume that the reviewer here meant Fig. 6]. AMP levels for starved cells are highest directly after the shift, and remain moderately high for the rest of*

the observed time. It would help to clarify how those patterns relate to the finding presented in Fig. 3E, which shows a gradually lowering in energy charge for the starved cells.

Answer: Please note that the reported ATP yield and the estimated maximal ATP production rates do not need to correlate with ATP, ADP or AMP concentrations as in general rates and metabolite levels do not need to correlate.

However, during thinking about this reviewer's comment and looking at our figures again, we notice that we unintentionally had reversed the time course in Figure 6 (in the heatmap figure), which now has been corrected. AMP levels for starved cells are actually highest at the later time points after the shift and are consistent with the changes in the adenylate energy charge shown in Fig. 3E.

9. Considering the schematic model presented in Fig. 7, it should be noted that ppGpp influences translation not only via the stringent response and the T/A-systems, but also directly through RMF (storing of deactivated ribosomes). ppGpp can thus be considered a regulator of the 'system-level' aspect presented here because an increase in ppGpp generally reduces growth rate (making fewer ribosomes available for translation).

Answer: We have modified the scheme to include this point.

10. It is unclear if the landscape in Fig. 8 is the result of a quantitative model or a schematic. Does the phenotypic instability (Z-axis) quantify the fractions persisters vs. growing cells in the whole population?

Answer: The landscape is just a schematic that conceptualizes our generated insights. The earlier used Z-axis description was indeed not overly clear. We now changed it.

Reviewer #2

The work presented here is a continuation of the elegant work by Kotte et al. 2014 where the Authors used the same experimental set-up to analyse phenotypic bistability during sudden carbon source shifts. Here they address the question of bacterial persistence induced by sudden carbon source shifts. The manuscript presents a huge amount of very carefully obtained data. However the logic, in particular, in the last part of Results, is difficult to follow. Most importantly, the Authors invoke ppGpp and SpoT in their model without presenting data to support their claim. They show strong data proposing that the sigma starvation factor RpoS is central to generating persisters. Fig 7A (the model) must be modified to reflect this fact or data supporting the involvement of SpoT should be added. In the same vein, the Authors seemingly use stringent response (mediated by ppGpp) and the general stress response (mediated by RpoS) as being one and the same but this is not correct (see e.g. reviews by Susan Gottesman and or Regine Hengge) since even though the two stress response regulons overlap they certainly are yet very different. Another general criticism is that the Results section contains a lot of interpretations. This is confusing and should be removed. Some of these passages are mentioned below.

Answer: Thanks a lot for the kind words about our work and the very constructive criticism. As outlined in the following, we carefully addressed these points in the revised manuscript.

Specific points:

The title is too general and in my mind could invoke the finding that persister formation depends strongly on RpoS (Fig 7B, C). Alternatively "Bacterial persistence" could be replaced by "Responsive diversification" in the title.

Answer: We added σ^S in the title of the manuscript to make it more specific.

Introduction:

28-29: reformulate sentence

Answer: We changed this sentence.

47: *Bacteria in macrophage vacuoles are starving and induce the stringent response to survive the starvation.*

Answer: We now make clear that persisters in a host can also be starved.

50-51: *sentence is imprecise.*

Answer: We clarified the sentence.

55-57 (important point): *It is well-described in the literature that persisters are heterogeneous, see e.g. Amato & Brynildsen, Current Biol 2015. However, the papers by Maisonneuve et al., 2013 and Nguyen et al., 2011 show that the vast majority of persisters (up to 99% of persisters in growing populations) depend on ppGpp in both organisms and also on TAS in E. coli. Where the minor fraction (less than 1%) of persisters come from is difficult to analyse because the numbers are so low. Such persisters are generated by another yet-to-be-discovered mechanism and are probably not highly relevant in the present context. The text should be rephrased to reflect this fact.*

Answer: We agree that persisters are heterogeneous – something that we also see in our antibiotic tolerance data. We now mention the point of heterogeneity in persisters explicitly in the re-written introduction.

64: *the authors mention Amato & Brynildsen 2014 and Kotte et al., 2014 in the same sentence. However, the experimental models used by the two groups are very different. In the diauxic growth model used by Amato & Brynildsen 2014, the cells gradually shift from one carbon source to another, a situation that often occurs in Nature. The experimental approach used here and also by Kotte et al., 2014 invokes a sudden shift from one carbon source to another and in my mind it is questionable whether bacteria encounter such conditions in more realistic settings. Nevertheless, using this "abrupt shift model" Kotte et al., 2014 elegantly showed that abrupt shifts from glucose to gluconeogenic C-sources induce "responsive diversification" that depended on the transcription factor Cra, an obvious and important result. Moreover the proteome resemblance and energy differences of starved and down-shifted cells certainly argue that the model is physiologically relevant and useful. However, it might be appropriate to state that the multidrug tolerant persisters generated by sudden metabolic shifts are useful experimental approximations to more realistic settings as also argued by researchers using stationary phase cells for the same purpose.*

Answer: In our opinion, both experimental models - i.e. the one used by us and the one used by Brynildsen – are probably still somewhat artificial experimental models. One could argue for either model (gradual and sudden) to occur more or less likely in nature. But as the reviewer also stated, this point is actually not so important. Importantly, however, according to the conceptual model that we present in the paper, both models (shifts) (and even the starvation model) are likely just different flavors of the same thing, namely a perturbation of metabolic homeostasis (with the difference just lying in the strength or rapidity of the perturbation). As such, we consider the different experimental models actually the same thing, and thus do not see a great need to divide things. Nevertheless, we added a statement to the respective sentence in the introduction highlighting that the applied shifts in these two papers were different.

Results:

Fig. S1A is very difficult if not impossible to understand. I had to go back to Fig 2C of Kotte et al., 2014 to understand what's going on. This makes the reading quite heavy.

Answer: We much better annotated and described the figure, and we hope that it is now easier to understand.

97: *"tolerance" should be "tolerant".*

Answer: Changed.

112-116: *The sentence is incomprehensible and should be clarified and moved to the Discussion since it invokes an interpretation.*

Answer: We agree that this sentence was really incomprehensible and changed it.

However, we opted to leave it where it was, because the finding suggested something that in a later point in the result section is picked up again. We feel that by this, the reader will be able to make better connections between the different points presented while going through the result part, in contrast to just connecting things at the end in the discussion. We note that it is maybe a bit a matter of taste, in how far results and discussion are intertwined. We hope that the reviewer will allow us to keep this short interpretation sentence (clearly highlighted as such) where it is.

141-142: "very similar" is unclear. Please explain.

Answer: We now specify in the respective sentence, which specific aspects we found to be similar. Also, we removed the qualifier "very".

294; "on" should be "or".

Answer: Done.

292-307 is a mix of results and interpretations and is slightly confusion. Delete or move to Discussion.

Answer: Again, at this point, we quickly hint to what this particular finding could indicate. In fact, we here just "discuss" a minor aspect. In the discussion at the end, we just discuss the global aspects of our work.

349-351: Here is an inference that pops up out of the blue. The authors write: "As RpoS was up-regulated for the whole period of our observation (Sup. Tab 5), this suggested that the stringent response is sustained over time,". Here the authors again make the inaccuracy of equalizing the RopS dependent general stress response and the stringent response. There are no data in the manuscript supporting that "the stringent response is sustained over time" and it should either be substantiated or removed. See also my comment to the model in Fig 7A below.

Answer: We are very grateful for pinpointing this inaccuracy. In the whole manuscript, we now correctly refer to general stress response, when we talk about rpoS. Furthermore, we now generated data on ppGpp levels, which we also added to the manuscript, with which we now can show that indeed also stringent response is activated in our persisters.

374: here the authors suddenly discuss similarities of starvation patterns of pro- and eukaryotes.

Answer: We removed the respective sentence and reference.

375: same as above, no data to support the claim about the stringent response.

Answer: We now measured the ppGpp levels and found that they are also increased in the persister cells compared to normally growing cells. We now show this data in Fig. 1C.

Fig 7A, the model. The article does not contain any data on ppGpp or SpoT and the model is therefore not supported by the data. SpoT and ppGpp should be removed or data to support the claim should be added.

Answer: As mentioned, we now quantified ppGpp and showed that it is elevated in persisters cells. Along with the lack of a phenotype in the RelA mutant, this implies that SpoT is responsible in controlling the ppGpp levels in our model system. Unfortunately, a SpoT mutant cannot be directly analyzed, as a SpoT mutant requires attenuated RelA activity and does not grow without amino acids present in the medium. We think that this additional data now better supports our model, in which we included an involvement of SpoT.

440: The authors are too rigorous in the interpretation of their own data. Fig 7B certainly shows that RpoS is highly important for the generation of persisters. It is a well-known fact that persisters

are generated by multiple parallel mechanisms so it is not to be expected that a single gene or pathway will account for all persisters in a given situation.

Answer: We agree that RpoS knockout has a tremendous effect on the obtained fraction of persisters, while still, 90% of cells enter persistence without RpoS action, even in absence of TAS. We rephrased our interpretation of the data, indicating that it is important.

I suggest the authors focus the paper more on e.g. RpoS whose expression, of course, is under the control of ppGpp and thus the stringent response. But to postulate that persister formation in their model is controlled by SpoT and ppGpp (which might certainly be true) needs experimental support.

Answer: For this revised version, we now measured the ppGpp levels in the persisters. Together with our finding that RelA deletion does not show a different phenotype (i.e no altered persister fraction), the increased ppGpp levels imply that SpoT must be responsible for the RpoS mediated stress response. We are thankful that the reviewer motivated us to do this long planned (but never done) experiment to determine the ppGpp levels.

Based on these considerations I suggest that the manuscript should be returned for a "major revision" allowing for either substantial rewriting and modification of the model (Fig 7A) or addition of crucial experiments showing the involvement of SpoT and ppGpp in the formation of persisters in the model used by the authors.

Answer: We now performed additional experiments showing the involvement of ppGpp in the formation of our persisters.

Reviewer #3:

The work of Radzikowski et al. compares the metabolism of non-growing cells mainly in two different conditions:

- 1. Shift from glucose to fumarate that results in growth arrest of most of the population and growth of a minority*
- 2. Shift to no carbon source medium: cells are abruptly starved and do not grow*

As expected, both conditions results in slower growth or growth arrest and therefore high tolerance of the whole population to several antibiotic drugs. Not surprisingly, both non-growing states are similar also in their expression levels of several TA systems and differ from glucose growing cells. Interestingly, they do find two drugs (CCCP and ofloxacin) that result in higher tolerance of the fumarate cells than the starved cells. The authors conclude that tolerance in the fumarate culture differs from tolerance due to starvation.

Focusing on the tolerance induced by the shift to fumarate, the authors show that these persisters are metabolically active and have a distinct metabolism when compared to cells growing on fumarate. By measuring the rate of fumarate uptake compared to the rate of oxygen uptake, they conclude that the ATP production is more efficient in persisters.

The core of the results in this work comes mainly from the extensive proteomic analyses of cells in various conditions:

Persisters cells as defined above (shift glucose to fumarate), starved cells (shift glucose to null, fumarate to null)), exponential fumarate, exponential glucose. This extensive data set enables the authors to see a general trend upon starvation that is followed upon all transitions within the PCA space, probably marking the main effect of growth reduction.

The authors then turn on specific proteins to understand what characterizes the proteome of persistent cells compared to starved cells and conclude that sigmaS is the major factor shaping the persister proteome. They categorize the proteins that are characteristic of the different states.

Finally, they show that rpoS is essential for triggering the growth arrest, whereas deletion of 10 TA modules has little effect.

This extensive analysis is then summarized in a new view of how persistence may be understood. The results presented here should be extremely interesting to microbiologists interested in the physiology of bacteria outside the exponential growth. The analysis includes here the transitions between carefully chosen states and their time-dependence and should shed light on the physiology on non-growing bacteria and their resilience to antibiotic treatment.

Minor comments:

- In Fig. 2A Cell size evaluation was done by super resolution microscopy and image analysis: it would be good to add images showing the typical differences in cell size at different time-points.

Answer: We have added the requested images to the figure.

- In Fig. 2C-E the legend states the persisters cells are measured but it is not clear what is plotted. What are the different grey shaded points? Are they still representing the two different conditions as in B?

Answer: The points were semi-transparent and this resulted in different shades of grey. We have modified the figure to avoid it. We also made the figure caption clearer.

- How can the authors evaluate the contribution of the 1% of fumarate growing cells to metabolism of the whole culture? How can the metabolism be attributed to persisters?

Answer: In the Appendix Text S1, we show an analysis that we did to demonstrate that the small fraction of growing cells does not significantly influence the reported values describing the phenotype of persisters. We realize that the earlier presented description of this analysis was probably too sparse. In this revised version, we now improved the respective text.

- The measured rate of oxygen and fumarate uptake do not show a steady state behavior. The graphs start increasing only towards the end. How can rates be extracted and steady-state analysis be done?

Answer: Yes, the reviewer is right. The cells were not in a full steady-state. The rates were extracted as predictions of a generalized additive model fitted to the data, which is somewhat similar to fitting a smoothing spline. For the FBA analysis, we assumed that these mean values represent steady-state value (similar to what was done in previous work). We have made this point clearer in the manuscript.

- in p. 10 the authors write that "the sum of adenylate nucleotides concentration is constant in persisters (Fig. 3F) but there is a 50% decrease in Fig. 3F."

Answer: In the process of a further statistical analysis of these data (in which we were trying to see whether the decrease is statistically significant), we realized that the earlier reported values were the values as they were present in the sample for the mass spectrometric analysis, and not the ones in the cell (as it should have been). The now correct data show that the changes in the sum of adenylate nucleotides changes in persisters and in starved cells are identical, and thus we concluded that the maintained AEC in persisters (in contrast to starved cells), must be due to energy generation. We apologize for this mistake.

- Fig 4 legend: it is written that the distance between points is inversely correlated with similarity between proteomes.

Answer: Yes, this is correct. The more similar the proteomes (and the higher Pearson's r), the lower the distance between the points. We made the figure legend more clear.

- Fig. 4 legend: Just to make the legend clearer: "circle" is usually for an open symbol (as in geometry) whereas "disk" is used for the filled symbol.

Answer: We have corrected the figure legends.

- The authors open the results with selected measurements of TA modules in persisters. How do these results concur with the proteome analyses? According to the little effect of the 10TAs deletion on their assay, it seems that TAs are not important in the type of persisters studied here.

Answer: Since the toxins are just small peptides or sRNAs, their detection with proteomics is difficult (or impossible) and the respective proteomics results are not very reliable. With the proteomics analysis we could only detect a few toxins. In fact, the lack of solid proteome data on TAS was the reason to perform the transcript abundance quantification. A comparison of these proteome data with the results from the RT-PCR measurements is not possible because we analyzed transcriptional activation of TA modules (no matter which the first gene of the operon was) because in most cases T/TA complex represses transcription of the operon, and only when they are activated (AT degraded) the promoter is de-repressed.

As for the second point on whether TAS are important or not. Apparently, they are not important in establishing the persister state under the conditions we look at, even though they have been found important in exponentially growing cultures. We do not think these observations are contradictory. Instead, both observations are in line with our notion that the state of persistence and the state of normal growth can be thought of as two attractors on a landscape divided by a watershed, where the two dimensions of this landscape are flux (on the x-axis) and the activity of growth-inhibiting mechanisms (for instance, TAS, on the y-axis) (cf. Fig 8): The key point is what type of perturbation is done, in which direction it moves the cell on the landscape and where the cell is initially located on the landscape. Apparently, in the nutrient-shift induced persisters, the metabolic flux perturbation dominates over the TAS effect.

- in p.21-22 the authors mention several times the "persistence level" of strains with and without *rpoS* changes but the figure shows only data on growth/no growth and not survival to antibiotics. As persistence is here a central theme, they should show survival assays of the mutants in Fig. 7B to one of the drugs tested in Fig. 1B.

Answer: Good point. We have performed antibiotic assays of the mutant strains and included the results in the appendix.

- The view of persistence and growth as two attractors of the metabolic fluxes is appealing. However, it is not clear what distinguishes "persistence" in this landscape from simply a growth/no growth landscape that is driven by starvation.

Answer: According to our findings that persister cells execute a global stress response (notably, a response that also starved cells attempt, but they ultimately fail to reach the stress protected state due to lacking resources), we think that persistence is more than just a split in growth/no-growth grow. We think that the mechanisms such as TAS, *rpoS* etc (the axis towards the back in the figure) help to actively push cells into the stress-protected state. It is a state that actually requires nutrients and can be only achieved with active metabolism. We have made this point clearer in the discussion.

We are now globally satisfied with the modifications made and I am pleased to inform you that we will be able to accept your paper for publication in Molecular Systems Biology pending a few minor amendments.

Corresponding Author Name: Matthias Heinemann

Manuscript Number: MSB-16-6998R